# From Indeterminacy to Determinacy: Augmenting Logical Reasoning Capabilities with Large Language Models

## Abstract

Recent advances in large language models (LLMs) have revolutionized the landscape of reasoning tasks. To enhance the capabilities of LLMs to emulate human reasoning, many prior works have focused on modeling intermediate reasoning steps using specific thought structures like chains, trees, or graphs. However, LLM-based reasoning continues to encounter challenges in three key aspects: 1) Selecting appropriate reasoning structures for various tasks; 2) Sufficiently and efficiently exploiting known conditions to deduce new insights; 3) Considering the impact of historical reasoning experience on future reasoning steps. To address these challenges, we propose DetermLR, a novel reasoning framework that formulates the reasoning process as a transformational journey from indeterminate premises to determinate ones. This process is marked by the incremental accumulation of determinate premises, making the conclusion progressively closer to clarity. DetermLR includes three essential components: 1) Premise identification: We categorize premises into two distinct types: determinate and indeterminate. This empowers LLMs to flexibly customize reasoning structures to match the specific task complexities. 2) Premise prioritization and exploration: We leverage quantitative measurements to assess the relevance of each premise to the target, prioritizing more relevant premises for exploring new insights. 3) Iterative process with reasoning memory: We introduce a reasoning memory module to automate storage and extraction of available premises and reasoning paths, preserving historical reasoning details for more accurate premise prioritization and exploration during iterative reasoning. Comprehensive experimental results demonstrate that DetermLR outperforms all baselines on four challenging logical reasoning tasks: LogiQA, ProofWriter, FOLIO, and LogicalDeduction. Compared to previous multi-step reasoning methods, DetermLR can achieve better reasoning performance while requiring fewer visited states, highlighting its superior efficiency and effectiveness in tackling logical reasoning tasks.

## 1 Introduction

The emergence of large language models (LLMs) has instigated a transformative wave within the realm of artificial intelligence (Zhao et al., 2023). The series models of GPT (Brown et al., 2020; Ouyang et al., 2022; OpenAI, 2023) and PaLM (Chowdhery et al., 2022; Anil et al., 2023) have exhibited remarkable proficiencies in natural language reasoning, which significantly contribute to the advancement of research and applications of cognitive intelligence (Huang & Chang, 2022). However, even the current state-of-the-art (SOTA) LLMs still grape with a fundamental limitation: lack of human-like advanced reasoning skills to rationally analyze known conditions and draw conclusions (Arkoudas, 2023; Singh et al., 2023). This leaves a substantial gap between LLM-based reasoning and the cognitive process of human reasoning.

To alleviate this limitation, existing works employ enhanced prompt engineering techniques to guide LLMs in eliciting intermediate thinking steps as evidence to ensure reliable conclusions (Zhou et al., 2022; Khot et al., 2022; Wei et al., 2022; Kojima et al., 2022). More recent works have focused on introducing more intricate reasoning structures, such as multiple chains (Wang et al., 2022b), trees (Yao et al., 2023) or graphs (Lei et al., 2023; Besta et al., 2023), to tackle increasingly com-

plex reasoning tasks. However, LLM-based reasoning continues to encounter the following three challenges: 1) Appropriate selection of reasoning structures: Since the task complexity cannot be solely inferred from the problem context, relying on a certain predefined structure to solve a variety of reasoning problems may create deficiencies in reasoning effectiveness or efficiency (Yao et al., 2023; Lei et al., 2023). 2) Skillfully exploiting known conditions: The literature on human cognitive reasoning provides valuable insights and emphasizes the importance of integrating available information to make informed decisions (Schaeken et al., 1999; Evans, 2002; Baron, 2023). This motivates cumulative reasoning (CR) (Zhang et al., 2023), which uses LLMs to iteratively generate new propositions based on available premises. However, CR still cannot approach the human thinking process, as it relies on randomly sampling existing premises for combination without a clear reasoning direction. 3) Accounting for historical reasoning experience: Previous works (Wei et al., 2022; Yao et al., 2023) often ignore historical reasoning details, resulting in lacking necessary key information for future reasoning steps.

To address these challenges and augment LLMs to grasp more human-like advanced reasoning skills, we analyze that three key factors demand consideration: 1) More precise identification of the determinacy of available premises for effective formulation of the reasoning process; 2) Prioritization of available premises for efficient exploration of new information; 3) Thoughtful memorization of historical reasoning details to guide the direction of the current reasoning step.

To this end, we propose DetermLR, a novel reasoning framework to align LLM-based reasoning more closely resemble human thinking. First, we formulate the logical reasoning process as a transformational journey from indeterminate premises to determinate ones. Since premises exhibit varying propositional characteristics and associations with the conclusion, we initiate the reasoning process with premise identification to finely categorize premises into two distinct types: determinate and indeterminate. Determinate premises are defined as simple statements, which can definitively contribute to conclusion derivation. In contrast, indeterminate premises may contain complex rules governing the relationships between multiple propositions. Regardless of the problem complexity, the reasoning process always follows the continuous accumulation of determinate information, indicating the flexibility and adaptability of DetermLR in handling logical reasoning tasks.

Second, human reasoning often seeks a "breakingthrough" from known conditions to deduce new insights, suggesting that the priority of different premises should be distinguished. Therefore, we introduce quantitative measurements to enable premise prioritization. We first identify the most relevant determinate premise to the conclusion, and then screen supplementary premises that are likely to interact with this primary premise. This prioritization steers LLMs to exclude irrelevant premises, and focus on more pertinent information for exploring new insights.

Third, we introduce a reasoning memory module to automate storage and extraction of premises and reasoning paths, preventing the loss of historical reasoning details. It supports both retrospective and prospective reasoning during the iterative process. Retrospectively, it stores historical reasoning details for updating available information. Prospectively, it extracts previous reasoning experience into future steps, enhancing the accuracy of premise prioritization and exploration.

To verify the capacity of LLMs to engage in rigorous logical reasoning based solely on established conditions, without external professional knowledge or common sense, we conduct extensive experiments on four challenging logical reasoning tasks: LogiQA, ProofWriter, FOLIO, and LogicalDeduction. The experimental results show that DetermLR achieves SOTA reasoning effectiveness, coupled with superior efficiency of requiring fewer visited states than previous multi-step reasoning methods like ToT and CR. Notably, in more intricate tasks like LogiQA, DetermLR exhibits even more pronounced advancements, mirroring human-like reasoning skills to a greater extent.

Our technical contributions to the advancement of LLM-based reasoning are summarized as follows:

1) We propose a novel perspective that formulates the logical reasoning process as a transformational journey from indeterminate premises to determinate ones, guiding LLMs to flexibly adapt reasoning structures to match specific complexities of various reasoning tasks.

2) We employ quantitative measurements for premise prioritization and exploration, enabling LLMs to prioritize premises more conductive for exploring new insights and improving reasoning effectiveness and efficiency.

3) We introduce a reasoning memory module to automate storage and extraction of available premises and reasoning paths, ensuring the consideration of essential historical reasoning details during the iterative reasoning process.

## 2 RELATED WORK

**LLM-based logical reasoning.** Previous methods mainly enhance reasoning by eliciting intermediate steps like chain-of-thought (CoT) (Wei et al., 2022; Wang et al., 2022b) and least-to-most prompting (Zhou et al., 2022). Extending the CoT concept, which follows a left-to-right progression, more recent works model thoughts as trees or graphs (Yao et al., 2023; Hu et al., 2023; Besta et al., 2023; Lei et al., 2023) to face more complex problems. Selection-inference (Creswell et al., 2022) refine the reasoning process of CoT by decomposing it into two modules: selection and inference. Algorithm-of-Thoughts (Sel et al., 2023) navigate reasoning pathways as in-context examples with merely few queries. Cumulative reasoning (Zhang et al., 2023) use higher-order logic rules for exploring new propositions based on given premises. Current LLM-based reasoning methods still face challenges in emulating human-like reasoning skills. In contrast, our method focuses on three key aspects: premise identification, premise prioritization and exploration, and iterative process with reasoning memory.

**Tasks for logical reasoning.** In the realm of logical reasoning tasks, various datasets have been utilized for evaluation (Khot et al., 2018; Wang et al., 2022a; Zhong et al., 2022; Nie et al., 2019; Bhagavatula et al., 2019; Welleck et al., 2018; Williams et al., 2017; Dagan et al., 2005; Bowman et al., 2015; Wang et al., 2018; Liu et al., 2021). LogiQA (Liu et al., 2020) involve various types of logical reasoning questions collected from the National Civil Servants Examination of China. ReClor (Yu et al., 2020), drawn from standardized graduate admission examinations, is used to examine logical reasoning skills. Based on Big-Bench (Srivastava et al., 2022) used to evaluate multi-aspect abilities of language models, Big-Bench Hard (BBH) (Suzgun et al., 2022) focuses on 23 challenging tasks for evaluating LLM-based reasoning. FOLIO (Han et al., 2022) is a human-annotated and logically complex datasets for natural language reasoning, equipped with first order logic (FOL) annotations. ProofWriter (Tafjord et al., 2020) is another commonly used dataset for deductive logical reasoning. Among these datasets, we carefully select four tasks of which premises are listed directly in the problem context and no need for additional summary.

## 3 METHOD

### 3.1 PROBLEM FORMULATION

The objective of a logical reasoning task can be regarded as using known conditionals and logical deduction rules to derive new essential intermediate information, culminating in an eventual target conclusion. Suppose a task provide a set of $N$ premises, denoted as $\mathcal{P} = (p_1, p_2, \cdots, p_N)$, the logical reasoning process can be formulated as:

$$c = \texttt{Reason}(p_1, p_2, \cdots, p_n), \tag{1}$$

where $c$ is the target conclusion of the task, and the mapping $\texttt{Reason}$ indicates how to use the given premises to derive the conclusion, which can be implemented based on instructing LLMs to understand the problem and provide new insights. Building upon the available premises and the target as input, we propose a novel reasoning framework to empower LLM-based reasoning to closely emulate the human cognitive reasoning. In the following sections, we will introduce in detail the following three pivotal components of the proposed method: 1) premise identification; 2) premise prioritization and exploration; 3) iterative process with reasoning memory.

### 3.2 PREMISE IDENTIFICATION

In the logical reasoning process, the premise set $\mathcal{P}$ stores determinate and indeterminate information for the conclusion $c$. The essential core of the reasoning process is to use logical rules to analyze and deduce premises, so that indeterminate information gradually evolves into determinate information. Therefore, we consider extending this consensus and propose a novel perspective for comprehending the essence of reasoning: **from indeterminacy to determinacy**.

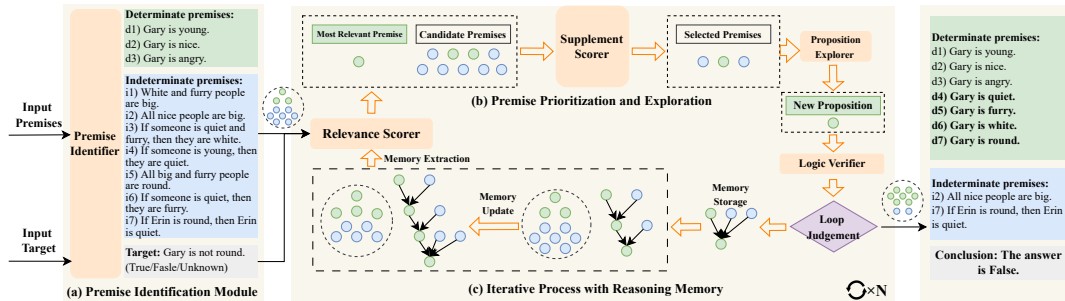

Figure 1: The framework overview of DetermLR.

To better simulate the reasoning process into determinate information accumulation, we develop a premise identification module to categorize input premises into two distinct types: determinate premises $\mathcal{D}$ and indeterminate premises $\mathcal{I}$. The criterion for premise identification is based on both the premise description itself and the input target $c$, which can be given by:

$$\mathcal{D}, \mathcal{I} = \texttt{Identify}(\mathcal{P}, c), \tag{2}$$

Specifically, we define determinate premises as simple statements that definitively lead to the desired conclusion. These premises state clarified facts or conditions, serving as the foundational blocks for reasoning. In contrast, indeterminate premises encompass propositions not directly related to the conclusion and often contain complex structures that reflect indeterminacy, such as disjunction ($x$ or $y$) and hypothesis (if $x$ then $y$). An indeterminate premise may be combined with other premises to establish a logical path to evolve into a determinate state.

As shown in Figure 1, the target revolves around "Gary" and "round", so simple statements including "Gary" are identified as determinate premises ($d_1 \cdots d_3$), while others are identified as indeterminate premises ($i_1 \cdots i_7$). Building upon this premise identification module, LLMs can eliminate the need for predefined structures and enhance the clarity and rigor of the reasoning process.

### 3.3 PREMISE PRIORITIZATION AND EXPLORATION

Once the original premises are categorized, how to better uncover the relationships between these premises to explore new insights is the next critical reasoning step. Prior sampling-based methods cannot distinguish the priority of different premises, leading to less skillful reasoning compared to human counterparts. Therefore, we aim to quantify the relationship between each premise and the conclusion, and prioritize premise selection to achieve more accurate and efficient reasoning.

**Premise prioritization.** We employ LLMs to implement two measurements `relevance` and `supplement` to select candidate premises. First, we score the relevance of each premise $p$ in conjunction with the conclusion $c$. By simulating the overlap of topics and elements in premises and the conclusion, it can help assign different priorities to premises throughout the reasoning process and give precedence to those more likely to lead to the conclusion. Second, we select the determinate premise $p_*$ that is most relevant to $c$ from $\mathcal{D}$ as the main reference. This mimics how humans seek key conditions as breakthroughs in the reasoning process to approach the conclusion. Once $p_*$ is determined, all other premises are considered candidate supplementary premises to interact with $p_*$. Thus, we quantify the likelihood of these premises being merged with $p_*$ according to semantic similarity and adherence to logical deduction rules. We can obtain supplementary premises with a given threshold $\theta$, denoted by $\mathbf{p}_s$. The two-stage scoring can be formulated as:

$$r_p = \texttt{relevance}(p, c), \quad p_* \triangleq \arg\max_{p \in \mathcal{D}} r_p \tag{3}$$

$$s_{p'} = \texttt{supplement}(p_*, p'), \quad \mathbf{p}_s \triangleq \{p' \in \mathcal{D} \cup \mathcal{I} \setminus \{p_*\}; s_{p'} \geq \theta\} \tag{4}$$

**Premise exploration.** Once candidate premises are determined, we employ LLMs to execute the `explore` function, which consider combining $\mathbf{p}_s$ with $p_*$ to generate a new proposition $\hat{p}$. Next, the rationality of the newly explored proposition $\hat{p}$ undergoes rigorous verification through our `verify`

function, encompassing three critical aspects: 1) Logical validity: We verify whether the deduction of the selected premises to $\widehat{p}$ is valid in term of logical reasoning rules; 2) Useful contribution: We verify whether $\widehat{p}$ is a useful determinate premise that contributes to derive the conclusion. It helps filter out the "correct nonsense" that may be logically valid but fail to enhance the conclusion derivation; 3) Duplication avoidance: We verify whether $\widehat{p}$ provides information gain beyond the original premises, avoiding the generation of mere paraphrases of existing premises. Only propositions that pass all these verification checks will be retained and added into the determinate premise set. The main steps of premise prioritization and exploration can be formulated as:

$$\widehat{p} = \texttt{explore}(p_*, \mathbf{p}_s), \quad \mathcal{D} := \mathcal{D} \cup \{\widehat{p}\}, \text{ if } \texttt{verify}(\widehat{p}, \{p_*, \mathbf{p}_s\}) = \text{True} \tag{5}$$

Through this process, LLMs can effectively harness more pertinent premises to explore determinate information, improving reasoning effectiveness and efficiency.

### 3.4 ITERATIVE PROCESS WITH REASONING MEMORY

To sustain the continuous progression of the reasoning process, the iterative exploration of new propositions becomes pivotal. Since known conditions are dynamically updated during the reasoning process, conventional methods often overlook historical reasoning details, leading to erroneous reasoning directions or stagnant reasoning progress (Yao et al., 2023; Zhang et al., 2023). In contrast, when humans tackle logical reasoning tasks, they usually record previous reasoning steps and retain both successful and unsuccessful attempts in mind. To bridge this cognitive gap, we design a reasoning memory module to automatically handle the storage and extraction of all available premises and evolving reasoning structures. The detailed operations of memory include:

**Memory storage.** During the $t$-th iteration of premise exploration, if the new proposition $\widehat{p}^{(t)}$ passes all verification checks, we store the updated set of determinate premises along with the reasoning path encompassing the links from original premises $\{p_*^{(t)}, \mathbf{p}_s^{(t)}\}$ to $\widehat{p}^{(t)}$ into the reasoning memory. Otherwise, we also store the combination $\{p_*^{(t)}, \mathbf{p}_s^{(t)}\}$ that failed the verification into memory as experience for subsequent reasoning steps.

**Memory extraction.** When we consider prioritizing premises in the $(t+1)$-th iteration, we extract $t$ previous reasoning steps from memory as experience to guide LLMs to avoid repeated errors. After each iteration of premise exploration, we also need to extract current premises and reasoning paths from memory, which can help accurately verify whether the current determinate information is sufficient to draw the conclusion.

Overall, the reasoning memory module supports both retrospective and prospective reasoning during the iterative process. Retrospectively, it stores historical reasoning details for updating available information. Prospectively, it extracts previous reasoning experience into future steps, enhancing the accuracy of premise prioritization and exploration.

## 4 EXPERIMENTS

To verify the capacity of LLMs to engage in rigorous logical reasoning based solely on established conditions, without external knowledge, we carefully select four challenging logical reasoning tasks: LogiQA, ProofWriter, FOLIO, and LogicalDeduction, and only reserve the problems in which premises are clearly delineated within the context for evaluation. Examples of each task are available in Table 1. Our proposed framework imposes no restrictions on the type of used LLMs. Here we uniformly use GPT-4 (OpenAI, 2023), the currently most advanced LLM as the engine to test the upper limit of LLM-based logical reasoning. Our implementation is based on the Microsoft guidance library [1]. We set the temperature to 0.1 by default and 0.7 for majority voting in CoT-SC.

Our comparisons include: 1) **Standard** prompting directly answers the question based on in-context examples; 2) **CoT** (Wei et al., 2022) adopts step-by-step generation of indeterminate rationales before the final answer; 3) **CoT-SC** (Wang et al., 2022b) uses majority voting to aggregate multiple CoTs; 4) **ToT** (Yao et al., 2023) models the reasoning process as a thought search tree; 5) **CR** (Zhang

---

[1]`https://github.com/guidance-ai/guidance`

Table 1: Logical reasoning examples of each task.

| Dataset | Problem | Original Premises | New Premises & Answer |
|---|---|---|---|
| LogiQA | There are 5 gardens of Pine, Bamboo, Plum, Orchid and Chrysanthemum. The east, south and north gates are located in three of these gardens respectively. The layout of gardens meets the conditions: 1) If the east gate is located on Pine or Chrysanthemum, then the south gate is not located on Bamboo; 2) If the south gate is not located in Bamboo, then the north gate is not located in Orchid; If the north gate is located in Orchid, which can be concluded? A. The south gate is located in Chrysanthemum. B. The east gate is located in Bamboo. C. The east gate is located in Plum. D. The east gate is located in Pine. | **Determinate premises:** d1) The north gate is located in Orchid. **Indeterminate premises:** i1) If the east gate is located on Pine or Chrysanthemum, then the south gate is not located on Bamboo; i2) If the south gate is not located in Bamboo, then the north gate is not located in Orchid; | d2) The south gate is not located in Bamboo. d3) The east gate is not located in Orchid, Bamboo, Pine or Chrysanthemum. d4) The east gate is located in Plum. The answer is C. |
| ProofWriter | Premises: 1) Metal conduct electricity. 2) Insulators do not conduct electricity. 3) If something is made of iron, then it is metal. 4) Nails are made of iron. Is the following statement true, false, or unknown? Nails cannot conduct electricity. | **Determinate premises:** d1) Metals conduct electricity. d2) Insulators do not conduct electricity. d3) Nails are made of iron. **Indeterminate premises:** i1) If something is made of iron, then it is metal. | d4) Nails are made of metal. d5) Nails conduct electricity. The answer is false. |
| FOLIO | Premises: 1) No giant language model has bad performance. 2) If a language model has good performance, it is used by researchers. 3) A work used by researchers should be popular. 4) If BERT is a giant language model, then the same for GPT3. 5) BERT is a giant language model. Is the following statement true, false, or unknown? GPT3 is popular. | **Determinate premises:** d1) A work used by researchers should be popular. **Indeterminate premises:** i1) No giant language model has bad performance. i2) If a language model has good performance, it is used by researchers. i3) If BERT is a giant language model, then the same for GPT3. i4) BERT is a giant language model. | d2) BERT is the same as GPT3. d3) GPT3 has good performance, it is used by some researchers. d4) GPT3 is popular. The statement is true. |
| LogicalDeduction | In an antique car show, there are three vehicles: a tractor, a convertible, and a minivan. The tractor is the second-newest. The minivan is newer than the convertible. Which of the following is true? A) The tractor is the oldest. B) The convertible is the oldest. C) The minivan is the oldest. | **Determinate premises:** d1) The tractor is the second-newest. **Indeterminate premises:** i1) The minivan is newer than the convertible. | d2) The minivan is newest. d3) The convertible is the oldest. The answer is C. |

et al., 2023) is recently proposed to generate new propositions based on available premises. We also compare several ablation variants of our DetermLR to test the impact of each component, including: 1) **DetermLR w/o identify** removes premise identification at the beginning of reasoning; 2) **DetermLR w/o priority** replaces premise priorities with randomly sampled candidate premises for exploration; 3) **DetermLR w/o memory** removes our memory module during iterative reasoning.

## 4.1 LOGIQA

**Task setup.** LogiQA (Liu et al., 2020) collects the multiple-choice logical problems from National Civil Servants Examination of China. This task aims to select the only correct option from four options based on the problem context. Since some questions rely on external common sense to be solved, we screened 179 questions with clear conditions from the test set to more objectively evaluate the logical reasoning capabilities of various methods based on given conditions.

Table 2: Comparison results on LogiQA.

| Method | Accuracy ↑ | # Visited States ↓ |
|---|---|---|
| Standard | 31.69% | 1 |
| CoT | 38.55% | 1 |
| CoT-SC | 40.43% | 16 |
| ToT | 43.02% | 19.87 |
| CR | 45.25% | 17 |
| DetermLR w/o identify | 46.15% | 17.24 |
| DetermLR w/o priority | 47.83% | 18.35 |
| DetermLR w/o memory | 39.66% | 11.98 |
| **DetermLR** | **54.19%** | **11.74** |

**Evaluation results.** The results in Table 2 show that all baselines including CR perform poorly on this task with accuracy below 46%. Since the utilization order of known conditions is crucial to solving the exam problem, baseline methods often fail to grasp the accurate reasoning direction. DetermLR can prioritize and memorize known conditions and improve accuracy to 54.19%. Meanwhile, the average number of visited states in DetermLR is 11.74, which is more efficient than CoT-SC, ToT and CR. Ablation results show that removing any component disrupts the reasoning process.

**Error analysis.** DetermLR achieves SOTA performance on this task, but it is still not comparable to humans. Current LLM-based reasoning cannot resolve the following errors: 1) Insufficient exploration of implicit conditions: LLMs cannot identify that school roommates have the same gender; 2) Insufficient understanding of boundary conditions: Three of the five are candidates, the first two and the last two each have one candidate, LLMs cannot assert that the middle one must be the candidate; 3) Lack of flexible use of logical rules: Given that A implies B, ¬ A implies B, LLMs cannot assert that B must be true.

## 4.2 PROOFWRITER

**Task setup.** ProofWriter (Tafjord et al., 2020) is a widely used logical reasoning benchmark. We use the open-world assumption subset where a question need to be proven true, false or unknown based on given statements. Since the dataset is divided into various reasoning depths, we follow Pan et al. (2023) to select 600 examples in the hardest depth-5 subset for evaluation.

Table 3: Comparison results on ProofWriter.

| Method | Accuracy ↑ | # Visited States ↓ |
|---|---|---|
| Standard | 46.83% | 1 |
| CoT | 67.41% | 1 |
| CoT-SC | 69.33% | 16 |
| ToT | 70.33% | 24.57 |
| CR | 71.67% | 16.76 |
| DetermLR w/o identify | 71.50% | 16.58 |
| DetermLR w/o priority | 72.32% | 17.21 |
| DetermLR w/o memory | 68.33% | 14.79 |
| **DetermLR** | **79.17%** | **14.63** |

**Evaluation results.** As shown in Table 3, DetermLR (79.17%) can make more accurate judgements on the target question than CoT-SC (69.33%), ToT (70.33%), and CR (71.67%). Also, DetermLR requires fewer visited states (14.63) to reach the same conclusion, ensuring more efficient premise integration and exploration. The performance of ablation variants degrades to similar to CR, indicating the necessity of these components in this task.

**Error analysis.** The average number of premises per case is over 15, and each case may contain many useless interference conditions. This makes it difficult to verify whether the new propositions are useful. LLMs may ignore some propositions that seem unrelated to the target, leading to deviations in the reasoning direction.

## 4.3 FOLIO

**Task setup.** FOLIO (Han et al., 2022) is a challenging benchmark for logical reasoning. The problems require complex first-order logic reasoning to solve. This task aims to determine whether a given statement is true, false, or unknown based on a set of premises. We follow the official data split and choose the validation set consisting of 204 examples for evaluation.

Table 4: Comparison results on FOLIO.

| Method | Accuracy ↑ | # Visited States ↓ |
|---|---|---|
| Standard | 60.29% | 1 |
| CoT | 67.65% | 1 |
| CoT-SC | 68.14% | 16 |
| ToT | 69.12% | 19.12 |
| CR | 69.11% | 15.87 |
| DetermLR w/o identify | 69.61% | 13.70 |
| DetermLR w/o priority | 70.59% | 14.69 |
| DetermLR w/o memory | 67.65% | 8.65 |
| **DetermLR** | **75.49%** | **8.57** |

**Evaluation results.** Table 4 shows that compared to all baseline methods, DetermLR can enhance reasoning accuracy (75.49%) and substantially reduce the number of visited states (8.57). Ablation results also demonstrate the importance of proposition identification, prioritization and exploration, and reasoning memory modules. Based on quantified prioritization, our required visited states are almost half of CR.

**Error analysis.** The average number of premises per case is lower than ProofWriter, but the descriptions of premises are often quite complex and indeterminate, making it challenging for LLMs to fully understand and effectively utilize the information for reasoning. For instance, consider the following premise: "If James is either a manager or in other countries, then James either has lunch at home and works remotely from home, or neither has lunch at home nor works remotely from home." The complexity hinders its utility for reasoning purposes.

## 4.4 LOGICALDEDUCTION

**Task setup.** LogicalDeduction (LD) is a challenging logical reasoning task in the BigBench benchmark (Srivastava et al., 2022). The problems are mainly about deducing the order of a sequence of objects from a set of conditions. We use the full test set consisting of 300 examples,

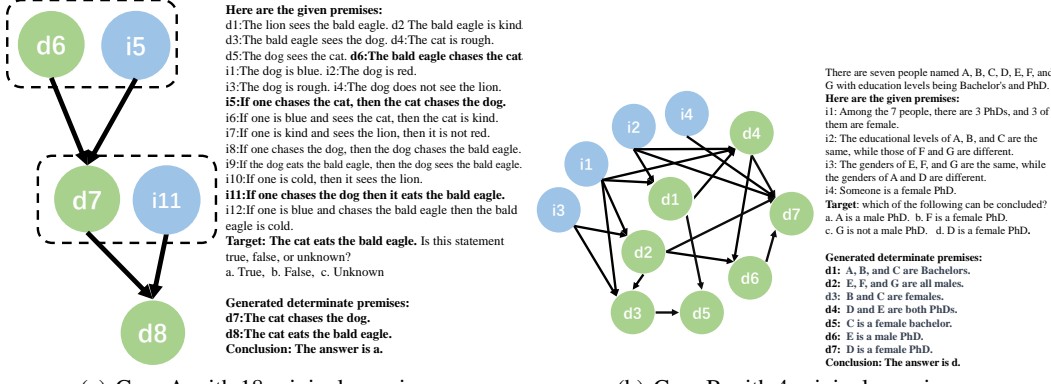

(a) Case A with 18 original premises.          (b) Case B with 4 original premises.

Figure 2: Two cases with contrasting reasoning structure and context complexity.

including 3,5,7 objects. In this task, we need to rigorously verify whether LLMs have used any other common sense knowledge apart from orientation.

**Evaluation results.** The results in Table 5 shows that compared to all baseline methods, DetermLR can enhance reasoning accuracy (85.00%) and substantially reduce the number of visited states (12.35). When memory is removed, accuracy decreases by 8.33%, demonstrating the importance of recording the inference structure and prompting LLMs to recall previously acquired information. Additionally, the number of visited nodes decreases from 17.02 to 12.35, indicating that prioritizing premise hierarchy can significantly improve reasoning efficiency.

Table 5: Comparison results on LogicalDeduction.

| Method | Accuracy ↑ | # Visited States ↓ |
|---|---|---|
| Standard | 71.33% | 1 |
| CoT | 73.33% | 1 |
| CoT-SC | 74.67% | 16 |
| ToT | 76.83% | 21.83 |
| CR | 78.33% | 16.98 |
| DetermLR w/o identify | 79.00% | 16.84 |
| DetermLR w/o priority | 80.33% | 17.02 |
| DetermLR w/o memory | 76.67% | 13.05 |
| **DetermLR** | **85.00%** | 12.35 |

**Error analysis.** In this task, a small number of repeated errors caused by relative position transformations. For instance, LLMs struggle to discern inconsistencies in boundary conditions that A is described as the fifth object from the left while B is stated as the third object from the right within a set of seven objects.

## 4.5 DISCUSSION

**Case study.** Based on human intuition, problems with a larger number of conditions and longer contexts tend to require more complex reasoning structures. We argue that predefined reasoning structure based on this experience may not always be correct. As shown in Figure 2, Case A presents 18 premises, initially appears to be a significantly complex problem compared to other tasks. However, upon prioritizing the premises, we discovered that the problem's reasoning could be modeled as a Chain-of-Thought (CoT) consisting of only two steps. Despite the problem's apparent simplicity, CoT cannot readily identify the aforementioned reasoning path within the vast array of premises. Meanwhile, Case B presents only 4 premises. But based on the results of DetermLR, it becomes evident that 7 determinate propositions were derived through iterative merging, resulting in the construction of a graphical reasoning structure. More reasoning examples and detailed reasoning processes are available in Appendix.

**Impact of number of determinate premises.** As shown in Figure 3, generating more number of specific propositions will streamline the reasoning process when addressing the question finally, resulting in fewer steps required and a decreased likelihood of errors made by LLMs. As the quantity

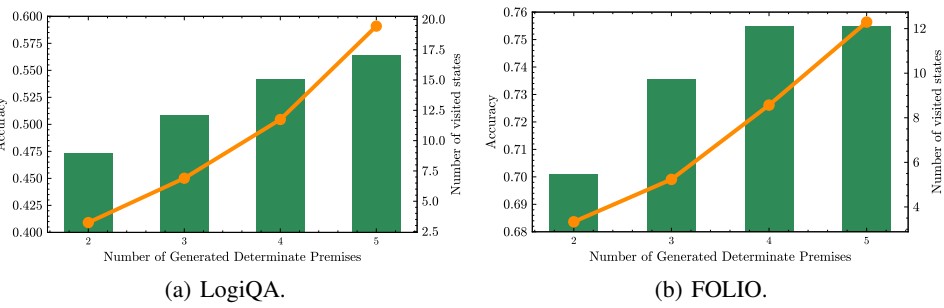

(a) LogiQA.                    (b) FOLIO.

Figure 3: The impact of number of generated determinate propositions.

of determinate propositions dwindles, the model's accuracy closely aligns with CoT-SC. Conversely, as the number of determinate propositions grows, the model's accuracy surpasses CR. But more propositions require more computing resources, for the following two reasons: 1) LLMs primarily focus on refining existing determinate propositions through format adjustments, like transforming "Billy is fat" into "If someone is Billy, then he is fat." 2) For complex datasets such as LogiQA, the number of premises implicitly dictates the quantity of propositions, rendering any attempt to derive additional propositions through extra loops futile.

**Generation efficiency of determinate premises.** When analyzing the average number of visited nodes required for each case, DetermLR and the baseline are quite close on some datasets, such as ProofWriter in Table 3. Nonetheless, this representation fails to provide an accurate reflection of actual performance of DetermLR. Consequently, we have computed the average number of visited states required to generate a correct proposition. As shown in Table 6, DetermLR outperforms the baseline significantly across all datasets. This also

Table 6: The average number of visited states per generated determinate premise.

| Method | LogiQA | ProofWriter | LD | FOLIO |
|---|---|---|---|---|
| ToT | 4.97 | 4.91 | 4.37 | 4.78 |
| CR | 4.25 | 3.35 | 3.40 | 3.97 |
| DetermLR w/o identify | 4.31 | 3.32 | 3.37 | 3.43 |
| DetermLR w/o priority | 4.59 | 3.44 | 3.40 | 3.67 |
| DetermLR w/o memory | 2.73 | 2.24 | 2.07 | 1.99 |
| **DetermLR** | **2.63** | **2.17** | **2.03** | **1.83** |

reflect the average difficulty of the datasets. For LogiQA, both CR (4.25) and DetermLR (2.63) are much higher than the values of other data sets on these two models, which reflects that the analytical reasoning questions in LogiQA are still the most challenging questions. To validate this hypothesis, we sought to obtain human performance on LogiQA. We invited two volunteers, one being a graduate student who had experience with public examinations, while the other was a student with no prior exposure to such knowledge. The former completed all the questions in 4 hours and achieved an accuracy rate of 73%, whereas the latter, with no time constraints, achieved an accuracy rate of 59%.

## 5 CONCLUSION

In this work, we propose DetermLR, a novel reasoning framework to align LLM-based reasoning more closely resemble human cognitive reasoning. First, we propose a novel perspective that formulates the reasoning process as a transition of indeterminate premises to determinate ones. Second, we employ quantitative measurements for premise prioritization and exploration, allowing LLMs to prioritize premises more conductive for exploring new insights. Furthermore, We introduce a reasoning memory module to automate storage and extraction of available premises and reasoning paths, ensuring the consideration of key historical reasoning details during the iterative reasoning process. Comprehensive experimental results show that DetermLR outperforms all baselines on four challenging logical reasoning tasks, while requiring fewer visited states, highlighting its superior efficiency and effectiveness in tackling logical reasoning tasks. Notably, in more intricate tasks like LogiQA, DetermLR exhibits even more pronounced advancements, mirroring human-like reasoning skills to a greater extent.

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

# A  MORE EXAMPLES ON DATASETS

## A.1  LOGICALDEDUCTION

---

**[An example in LogicalDeduction]**

**Context:**
The following paragraphs each describe a set of seven objects arranged in a fixed order. The statements are logically consistent within each paragraph.In a golf tournament, there were seven golfers: Ana, Eli, Mya, Amy, Joe, Mel, and Ada. Ada finished above Mel. Ada finished third-to-last. Amy finished above Ana. Mya finished second-to-last. Joe finished above Amy. Eli finished below Ana. Ana finished above Ada. Ana finished above Mya.

**Question:** Which of the following is true?

**Options:**
A) Ana finished third.
B) Eli finished third.
C) Mya finished third.
D) Amy finished third.
E) Joe finished third.
F) Mel finished third.
G) Ada finished third.

**Answer: A**

---

**[Chain-of-Thought Reasoning by GPT-4]**

**Conclusion:**

- From the premises, we know that: 1. Ada finished third-to-last, which means Ada finished 6th. 2. Mya finished second-to-last, which means Mya finished 6th. 3. Joe finished above Amy. 4. Amy finished above Ana. 5. Eli finished below Ana. 6. Ada finished above Mel. Now, let's try to arrange the golfers based on the given information: 1. Joe 2. Ana 3. Amy 4. Eli 5. - 6. Mya and Ada 7. Mel From the arrangement, we can see that Amy finished third. So, the answer is D) Amy finished third.

**[Prediction: D]** (Wrong)

---

**[Cumulative Reasoning by GPT-4]**

**Generated Propositions:**

- Mel finished last.
- Ana finished fifth.
- Eli finished sixth.
- Mel finished last. Ada finished fifth.
- Amy finished above Ana and below Joe.
- Joe finished third.

**Conclusion:**

- From the propositions, we know that Joe finished third. So the answer is E.

**[Prediction: E]** (Wrong)

---

**[DetermLR by GPT-4]**

**Generated Determinate Premises:**

- Mya did not finish first.
- Ada did not finish first or second.
- Joe finished above Ana.
- Amy did not finish third-to-last.
- Amy finished above Eli.
- Amy finished second. Joe finished first.
- Ada did not finish first-to-third.
- Eli did not finish first-to-third.

**Reasoning Details:**

1. In the NO:1 round, we use these "premises": "Amy finished above Ana. Mya finished second-to-last. Ada finished above Mel. Eli finished below Ana. Joe finished above Amy." and got a "New Determinate Premise": "Joe finished above Ana"

2. In the NO:2 round, we use these "premises": " Amy finished above Ana. Eli finished below Ana." and got a "New Determinate Premise": "Amy finished above Eli."

3. In the NO:3 round, we use these "premises": " Ada finished third-to-last. Amy finished above Ana." and got a "New Determinate Premise": "Amy did not finish third-to-last."

4. In the NO:4 round, we use these "premises": "Amy finished first or second. Joe finished above Amy. " and got a "New Determinate Premise": "Amy finished second. Joe finished first."

5. In the NO:5 round, we use these "premises": " Ada finished third-to-last. Ada finished above Mel. " and got a "New Determinate Premise": "Ada did not finish first-to-third. "

6. In the NO:6 round, we use these "premises": " Eli finished below Ana. Ana did not finish first or second. " and got a "New Determinate Premise": "Eli did not finish first-to-third. "

**Conclusion:**

- From the premises and propositions, we know that Ada finished third-to-last, Ana finished above Ada, Ana finished above Mya, which means she finished fifth. Mya finished second-to-last, which means she finished sixth. Since there are only seven golfers, this means that Joe must have finished first, as he finished above Amy, who finished above Ana, who finished above Eli. With Joe finishing first, Amy must have finished second, since she finished above Ana and Eli. Ana would be third, Eli would be fourth, and Mel would be the last (seventh). So the answer is A) Ana finished third.

**[Prediction: A]** (Right)

## A.2 LOGIQA

---

**[An example in LogiQA]**

**Context:**
There are seven people named A, B, C, D, E, F, and G with education levels being Bachelor's and PhD. Here are the given premises: 1): Among the 7 individuals, there are 3 with a PhD., and 3 of them are female. 2) The educational levels of A, B, and C are the same, while those of F and G are different. 3) The genders of E, F, and G are the same, while the genders of A and D are different. 4) The final hire is a female PhD.

**Question:** Based on the statements above, which of the following can be concluded?

**Options:**
A. A is a male PhD.
B. F is a female PhD.
C. G is not a male PhD.
D. D is a female PhD.

**Answer: D**

---

**[Chain-of-Thought Reasoning by GPT-4]**

**Conclusion:**

- From the premise, it can be inferred that the genders of E, F, and G are female. Since the final appointment is a female Ph.D. holder, the selection must be made from E, F, and G. The educational levels of A, B, and D are the same, so they cannot be Ph.D. holders because there are only 3 Ph.D. holders. Therefore, there must be two Ph.D. holders among E, F, and G. Since the educational levels of F and G are different, one of them must be a Ph.D. holder, and E is also a Ph.D. holder. Therefore, one of E, F, and G is a female Ph.D. holder. The genders of A and C are different, and since E, F, and G are females, either A or C must be male. Among options A, B, C, and D, option B is correct because F is a female Ph.D. holder. The answer is B.

**[Prediction: B]** (Wrong)

---

**[Cumulative Reasoning by GPT-4]**
**Generated Propositions**

- The genders of D,E,F are female.
- A,B,C are undergraduates.

**Conclusion:**

- From the proposition, we can deduce that E, F, and G are females. A and C have different genders, so A and C are males. A, B, and D have the same educational background, which means they are either all Ph.D. holders or all undergraduates. However, there are only 3 Ph.D. holders, so A, B, and D must all be undergraduates. This means that E, F, and G include two Ph.D. holders. Since the final appointment is a female Ph.D. holder, it means that either E or F must be a Ph.D. holder. However, the educational levels of F and G are different, so G is an undergraduate. Therefore, both E and F are Ph.D. holders. Therefore, the final answer is: A is a male undergraduate; C is a male undergraduate; E is a female Ph.D. holder; F is a female Ph.D. holder; G is a female undergraduate. The answer is C.

**[Prediction: C]** (Wrong)

---

**[DetermLR by GPT-4]**

**Generated Determinate Premises:**

- A, B, and C have Bachelor's degrees.
- E, F, and G are all males.
- B and C are females.
- D and E are both PhD. holders.
- C is a female undergraduate.
- E is a male PhD. holder.
- D is a female PhD. holder.

**Reasoning Details:**

1. In the NO:1 round, we use these "premises": "Among the 7 individuals, there are 3 with a PhD., and 3 of them are female. The educational levels of A, B, and C are the same, while those of F and G are different." and got a "New Determinate Premise": "A, B, and C have Bachelor's degrees."

2. In the NO:2 round, we use these "premises": " Among the 7 individuals, there are 3 with a PhD., and 3 of them are female. The genders of E, F, and G are the same, while the genders of A and D are different." and got a "New Determinate Premise": "E, F, and G are all males."

3. In the NO:3 round, we use these "premises": " Among the 7 individuals, there are 3 with a PhD., and 3 of them are female. The genders of E, F, and G are the same, while the genders of A and D are different. E, F, and G are all males" and got a "New Determinate Premise": "B and C are females."

4. In the NO:4 round, we use these "premises": "Among the 7 individuals, there are 3 with a PhD., and 3 of them are female. The genders of E, F, and G are the same, while the genders of A and D are different. A, B, and C have Bachelor's degrees." and got a "New Determinate Premise": "D and E are both PhD. holders."

5. In the NO:5 round, we use these "premises": "B and C are females. A, B, and C have Bachelor's degrees." and got a "New Determinate Premise": "C is a female undergraduate."

6. In the NO:6 round, we use these "premises":" D and E are both PhD. Holders. E, F, and G are all males " and got a "New Determinate Premise": "E is a male PhD. holder." In the NO:7 round, we use these "premises": "The educational levels of A, B, and C are the same, while those of F and G are different. The final hire is a female PhD. E is a male PhD. holder. "and got a "New Determinate Premise": " D is a female PhD. holder"

**Conclusion:**

- From the propositions, we know the answer is D. D is a female PhD.

**[Prediction: D]** (Right)

A.3 PROOFWRITER

---

**[An example in ProofWriter]**

**Context:**

The bald eagle chases the cat. The bald eagle is kind. The bald eagle sees the dog. The cat is rough. The dog is blue. The dog is red. The dog is rough. The dog sees the cat. The dog does not see the lion. The lion sees the bald eagle. If something chases the cat then the cat chases the dog. If something chases the dog then it eats the bald eagle. If something is kind and it sees the lion then it is not red. If something chases the dog then the dog chases the bald eagle. If the dog eats the bald eagle then the dog sees the bald eagle. If something is cold then it sees the lion. If something sees the cat and it is blue then the cat is kind. If something is blue and it chases the bald eagle then the bald eagle is cold.

**Question:** Based on the above information, is the following statement true, false, or uncertain? The cat eats the bald eagle.

**Options:**
A) True
B) False
C) Uncertain

**Answer: A**

---

**[Chain-of-Thought Reasoning by GPT-4]**

**Conclusion:**

- The bald eagle chases the cat.2. If something chases the cat, then the cat chases the dog. (From 1, the cat chases the dog)3. If something chases the dog, then it eats the bald eagle. (We don't have information about something chasing the dog) 4. The cat eats the bald eagle. (We don't have information about the cat eating the bald eagle)Recall the Hypothesis: The cat eats the bald eagle is Unknown.

**[Prediction: C]** (Wrong)

---

**[Cumulative Reasoning by GPT-4]**

**Generated Propositions:**

- The bald eagle does not eat the dog.
- If the bald eagle sees the lion, it is not red.
- The bald eagle is not red.
- If the dog chases the cat, then the cat chases the dog.
- The dog does not eat the bald eagle.

**Conclusion:**

- We can not find any direct or indirect information about the cat eating the bald eagle. The information provided in the premises does not support or contradict the hypothesis. Judgement: Now we know that the Hypothesis is Unknown.

**[Prediction: C]** (Wrong)

---

**[DetermLR Reasoning by GPT-4]**

**Generated Determinate Premises:**

1. The cat chases the dog.
2. The dog chases the bald eagle.
3. Something chases the cat.
4. The bald eagle is something that chases the cat.
5. The dog and the lion both see the bald eagle.
6. If the cat chases the dog, then the cat eats the bald eagle.
7. The cat eats the bald eagle.

**Reasoning Details:**

- In the NO:1 round, we use these "premises": "If something chases the cat then the cat chases the dog. The bald eagle chases the cat."and got a "New Determinate Premise": "The cat chases the dog."

- In the NO:2 round, we use these "premise": "If something chases the dog then the dog chases the bald eagle. If something chases the cat then the cat chases the dog. The dog sees the cat. The cat chases the dog. If something chases the dog then it eats the bald eagle. The bald eagle chases the cat. The cat is rough."and got a "New Determinate Premise": "The dog chases the bald eagle."

- In the NO:3 round, we use these "premises": "If something chases the cat then the cat chases the dog. The cat chases the dog. If something chases the dog then it eats the bald eagle. The bald eagle chases the cat."and got a "New Determinate Premise": "Something chases the cat."

- In the NO:4 round, we use these "premises": "Something chases the cat. The bald eagle chases the cat."and got a "New Determinate Premise": "The bald eagle is something that chases the cat."

- In the NO:5 round, we use these "premises": "The dog chases the bald eagle. The bald eagle is something that chases the cat. The bald eagle chases the cat. The bald eagle is kind. The bald eagle sees the dog. The lion sees the bald eagle."and got a "New Determinate Premise": "The dog and the lion both see the bald eagle."

- In the NO:6 round, we use these "premises": "The dog chases the bald eagle. The bald eagle sees the dog. If something chases the dog then the dog chases the bald eagle. If something chases the dog then it eats the bald eagle. The dog is blue." and got a "New Determinate Premise": "If the cat chases the dog, then the cat eats the bald eagle."

- In the NO:7 round, we use these "premises": "The cat chases the dog. The bald eagle is something that chases the cat. If something chases the dog then it eats the bald eagle. If something chases the cat then the cat chases the dog. If the cat chases the dog, then the cat eats the bald eagle. The dog chases the bald eagle. The bald eagle chases the cat."and got a "New Determinate Premise": "The cat eats the bald eagle."

**Conclusion:**

- From the propositions, we know the cat eats the bald eagle. So the answer is true.

**[Prediction: A]** (Right)

## A.4 FOLIO

---

**[An example in FOLIO]**

**Context:**
No digital media are analog. Every printed text is analog. All streaming services are digital. If an object is a hardcover book then it is printed text. If 1984 is a streaming service, then 1984 is a hardcover book.

**Question:** Based on the above information, is the following statement true, false, or uncertain? 1984 is not a streaming service.

**Options:**
A) True
B) False
C) Uncertain
**Answer: A**

---

**[Chain-of-Thought Reasoning by GPT-4]**

**Conclusion:**

- From the premises, we cannot directly deduce whether the hypothesis 1984 is not a streaming service is true or false. The premises provide information about the different media formats, but there is no explicit information about the format of 1984. So the answer is C.

**[Prediction: C]** (Wrong)

---

**[Cumulative Reasoning by GPT-4]**

**Generated Propositions:**

- If 1984 is a streaming service, then it is digital.
- If 1984 is a streaming service, then it is not analog.

**Conclusion:**

- The premises state that if 1984 is a streaming service, then it is a hardcover book. However, no information is provided about whether 1984 is a streaming service or a hardcover book. Since we lack information to confirm or deny the hypothesis, we cannot deduce a conclusion from the given premises. Judgement: Now we know that the Hypothesis is Unknown. So the answer is C.

**[Prediction: C]** (Wrong)

---

**[DetermLR Reasoning by GPT-4]**

**Generated Determinate Premises:**

- If 1984 is a streaming service, then 1984 is digital.
- All streaming services are not analog.
- If 1984 is a streaming service, then it cannot be a hardcover book.

**Reasoning Details:**

1. In the NO:1 round, we use these "premises": "If 1984 is a streaming service, then 1984 is a hardcover book. All streaming services are digital." and got a "New Determinate Premise": "If 1984 is a streaming service, then 1984 is digital."

2. In the NO:2 round, we use these "premises": "No digital media are analog. All streaming services are digital. If 1984 is a streaming service, then 1984 is a hardcover book. If 1984 is a streaming service, then 1984 is digital." and got a "New Determinate Premise": "All streaming services are not analog."

3. In the NO:3 round, we use these "premises": "If 1984 is a streaming service, then 1984 is digital. If 1984 is a streaming service, then 1984 is a hardcover book. All streaming services are digital. No digital media are analog. All streaming services are not analog." and got a "New Determinate Premise": "If 1984 is a streaming service, then it cannot be a hardcover book."

**Conclusion:**

- We know from the premises that if 1984 is a streaming service, then 1984 is a hardcover book. However, we also deduced that if 1984 is a streaming service, then it cannot be a hardcover book, as it would be digital and not analog. This contradiction leads us to conclude that 1984 is not a streaming service. Judgement: Now we know that the Hypothesis is True.

**[Prediction: A]** (Right)

## B    PROMPT TEMPLATES

These four selected logical reasoning tasks have different data formats. Among them, ProofWriter and FOLIO offer explicit known premises, while LogiQA and LogicalDeduction require parsing the question context to extract the premises, which indicates that extra steps such as premise transformation are necessary. We use ProofWriter and LogicalDeduction as representative tasks to illustrate the prompt templates. More details about prompt design are available in our Github repository.

### B.1    PROOFWRITER

Based on the modeling scheme introduced by our DetermLR, we summarize main designed prompts into several parts such as premise identification, premise prioritization, premise exploration, logical validation, and final conclusion.

---

**Prompts used for Premise Identification**
{{#system}}Suppose you are one of the greatest AI scientists, logicians and mathematicians. Let us think step by step.
First, read and analyze the following definition:
Determinate premise: The premise contains the same noun or adjective as the Hypothesis, and the premise is not in the structure of "if..." or "if...then...".
Second, read and analyze the "Premise" and "Hypothesis" .Judge "Premise" is "determinate premise" or not.
Third, please make sure your classification decisions are derived directly from definitions, rather than unsourced common sense.
—
{{system}}
{{#each examples}}
{{#user}}
—
"Premise": "{{this.Premise}}"
"Hypothesis": "{{this.Hypothesis}}"
{{/user}}
{{#assistant}}"Judgement":"Is this "Premise" a "determinate premise" or not?
{{this.usefulness}}" {{/assistant}}
{{#assistant}}"Explanation": {{this.Explanation}}{{assistant}}
{{/each}}

---

**Prompts used for Premise Prioritization**

{{#system}}Suppose you are one of the greatest AI scientists, logicians and mathematicians. Let us think step by step. Read and analyze the "determinate premise" and "indeterminate premise"

first, then selecting several premises from them.

Read the "Last reasoning history".If we got a "false Proposition" in history,when you select "Most relevant premise",do not choose the same "Most relevant premise" in history as your answer.

Please follow these steps:

1.From the determinate premise, select the "Most relevant premise" which has the same subject with "Hypothesis", and give a score from 0 to 1.

2.You need to assess how the "Most relevant premise" relates to all the other "determinate premise" and "indeterminate premise",based on Relevance scoring rules.

3.The "determinate premise" and "indeterminate premise" with scores higher than 0.25 will be used as the final results, along with Most relevant premise.

Relevance scoring rules:

1. When scoring relevance, 0.25 added for each noun or 0.3 added for each adjective that is the same between two sentences.

2. Scores start to accumulate from 0 points, and the upper limit is 1 point.

3. If sentence p1 is a hypothetical premise of sentence p2,then add 0.25 to p2. for example: measure "if A then B." and "A is true." Then add 0.25 points to "if A then B".

—-

{{system}}

{{#each examples}}

{{#user}}

—

"determinate premise": "{{this.determinate premise}}"

"indeterminate premise": "{{this.indeterminate premise}}"

"Hypothesis": "{{this.Hypothesis}}"

"Last reasoning history": "{{this.last history}}"

{{user}}

{{#assistant}}Can you select the premise from the "determinate premises" that scores the highest score for Relevance scoring rules to the "hypothesis"?{{assistant}}

{{#assistant}}"Most relevant premise": "{{this.Most relevant premise}}"{{assistant}}

{{#assistant}}Can you assess how the "Most relevant premise" relates to all the other "determinate premise" and "indeterminate premise" accoding to Relevance scoring rules?{{assistant}}

{{#assistant}}"Other premises scores": "{{this.Other premises scores}}"{{assistant}}

{{#assistant}}"Results": "{{this.Results}}"{{assistant}}

{{~each}}

---

**Prompts used for Premise Exploration**

{{#system}}Suppose you are one of the greatest AI scientists, logicians and mathematicians. Let us think step by step.

Please use Logical Reasoning Rules(LRR) to deduce a "Proposition" from two given "Premises" and the proposition does not include "if".

Logical Reasoning Rules(LRR):

1. "Two premises": "If A,then B. A is true." then "Proposition": "B is true."

2. "Two premises": "If A,then B. B is not true." then "Proposition": "A is not true"

3. "Two premises": "A is either C or D. A is not C." then "Proposition": "A is D."

Please make sure that the "Proposition" is logically correct.

Please make sure that the "Proposition" is not a duplicate of the "Premises".

Please make sure your reasoning is directly deduced from the "Premises" and "Propositions" other than introducing unsourced common knowledge and unsourced information by common sense reasoning.

Please remember that your "Proposition" should be useful to determine whether the "Hypothesis" is True, False or Unknown.

---
{{system}}

{{#each examples}}
{{#user}}
—
"Premises": "{{this.premises}}"
We want to deduce more propositions to determine the correctness of the following "Hypothesis":
"Hypothesis": "{{this.conclusion}}"
Can you deduce a new "Proposition" from at least two given "Premises"?
{{user}}

{{#assistant}}"Proposition": "{{this.proposition}}"{{assistant}}
{{~each}}
{{#user}}
—
"premises": "{{premises}}"
"boundary condition": "{{boundary condition}}"
We want to derive more propositions to solve the following question:
"question": "{{question}}"
Combined with boundary conditions, can you derive a new "proposition" from at least two given "premises"?
{{user}}

{{#assistant}}"proposition": "{{assistant}}"
{{#assistant}}{{gen "proposition" temperature=temperature max tokens=100 stop='
'}}{{assistant}}

---

**Prompts used for Logical Validation**
{{#system}}Suppose you are one of the greatest AI scientists, logicians and mathematicians. Let us think step by step.
Please use the Logical Reasoning Rules(LRR) to determine whether the deduction of the given "Premises" to a "Proposition" is valid or not, reply with True or False.
Logical Reasoning Rules(LRR):
1. "Two premises": "If A,then B. A is true." then "Proposition": "B is true."
2. "Two premises": "If A,then B. If B,then C." then "Proposition": "If A, then C."
3. "Two premises": "If A,then B. B is not true." then "Proposition": "A is not true"
4. "Two premises": "A is either C or D. A is not C." then "Proposition": "A is D."
—
{{system}}
{{#each examples}}
{{#user}}
—
"Premises": "{{this.premises}}"
"Proposition": "{{this.proposition}}"
{{user}}
{{#assistant}}"Judgement": "Is this deduction valid? {{this.validation}}"{{assistant}}
{{~each}}

---

**Prompts used for Final Conclusion**
{{#system}}Suppose you are one of the greatest AI scientists, logicians, and mathematicians. Let's think about it step by step.
First read and analyze the "paragraphs" and "questions", then use the "premises", "boundary

conditions" and "propositions" to reason which of the options given is the answer to the "question".

Make sure that your reasoning is derived directly from "premises" and "propositions" rather than introducing passive common sense and passive information through common sense reasoning.

Please note that this is a single choice question.

If you can get the answer directly from the proposition, then you should choose the answer directly, otherwise keep reasoning with the proposition, premises, and boundary conditions until you arrive at a single answer.

—

{{system}}
{{#each examples}}
{{#user}}

—

"context": "{{this.context}}"
"question and options": "{{this.question}}"
{{user}}
{{#assistant}}"Premises": "Let's think step by step, and from the context we can extract these premises: {{this.premises}}"{{assistant}}
{{#assistant}}"Boundary condition": "Let's think step by step, and from the context we can extract these boundary conditions: {{this.boundary condition}}"{{assistant}}
{{#assistant}}"Thoughts": "Let us think step by step. From the premises, we can deduce propositions:{{this.propositions}}"{{assistant}}
{{#assistant}}"Recall the questions and options":"{{this.question}}"{{assistant}}
{{#assistant}}"Reasoning": "Using premises, boundary conditions, and continuing to reason according to the propositions already obtained,{{this.reasoning}}"{{assistant}}
{{#assistant}}"Recall the questions and options":"{{this.question}}"{{assistant}}
{{#assistant}}"Judgement": "Now we know that the answer to this question should be{{this.ans}}"{{assistant}}
{{~each}}
{{#user}}

—

"context": "{{context}}"
"question and options": "{{question}}"
{{user}}
{{#assistant}}"Premises": "Let's think step by step, and from the context we can extract these premises: {{premises}}"{{assistant}}
{{#assistant}}"Boundary condition": "Let's think step by step, and from the context we can extract these boundary conditions: {{boundary condition}}"{{assistant}}
{{#assistant}}"Thoughts": "Let us think step by step. From the premises, we can deduce propositions:{{propositions}}"{{assistant}}
{{#assistant}}"Recall the reasoning history":"{{infer history}}"{{assistant}}
{{#assistant}}"Recall the questions and options":"{{question}}"{{assistant}}
{{#assistant}}"Reasoning": "Using premises, boundary conditions, and continuing to reason according to the propositions already obtained,{{assistant}}
{{#assistant}}{{gen "reasoning" temperature=0.7 max tokens=500 stop=[' ']}}{{assistant}}
{{#assistant}}"Recall the questions and options":"{{question}}"{{assistant}}
{{#assistant}}"Judgement": "Now we know that the answer to this question should be{{assistant}}
{{#assistant}}{{select "judgement" options=choose}}{{assistant}}

## B.2 LOGICALDEDUCTION

In addition to the prompting steps mentioned above, we also include premise extraction and premise transformation to parse the available premises from the original question.

---

**Prompts used for Premise Identification**

{{#system}}Suppose you are one of the greatest AI scientists, logicians and mathematicians. Let us think step by step.
First, read and analyze the following definition:
Determinate premise: The premise contains the same noun or adjective as the Hypothesis, and the premise is not in the structure of "if..." or "if...then...".
Second, read and analyze the "Premise" and "Hypothesis" .Judge "Premise" is "determinate premise" or not.
Third, please make sure your classification decisions are derived directly from definitions, rather than unsourced common sense.
—
{{system}}
{{#each examples}}
{{#user}}
—
"Premise": "{{this.Premise}}"
"Hypothesis": "{{this.Hypothesis}}"
{{/user}}
{{#assistant}}"Judgement":"Is this "Premise" a "determinate premise" or not? {{this.usefulness}}" {{/assistant}}
{{#assistant}}"Explanation": {{this.Explanation}}{{assistant}}
{{/each}}

---

**Prompts used for Premise Prioritization**

{{#system}}Suppose you are one of the greatest artificial intelligence scientists, logicians, and mathematicians. Let's think about it step by step.
First read and analyze the "determinate premises" and "indeterminate premises", and then filter out several premises.
When you decide on a variable, read through the inference history first and don't choose a variable that has failed before as your choice for this round.
Please follow these steps:
1. Count the cumulative number of times each variable is mentioned by "determinate premises" and "indeterminate premises".
2. Determine the variable according to the number of mentions from high to low. If the number of mentions is the same, the variable with more prerequisites will be given priority.
3. Determine whether the value of the variable has been determined under the current variable. If it is determined, search and determine the next variable in order from most to least. If it has not been completely determined, go to step 4.
4. Use this variable as a criterion for screening "premises" and filter out all premises related to this variable.
—
{{system}}
{{#each examples}}
{{#user}}
—
"determinate premise": "{{this.determinate premise}}"
"indeterminate premise": "{{this.indeterminate premise}}"
"topic": "{{this.topic}}"
"boundary condition": {{this.boundary condition}}
"Inference history": "{{this.last false history}}"
{{user}}
{{#assistant}}Can you count the cumulative number of times each variable is mentioned by

---

the premises?{{assistant}}
{{#assistant}}"Count": "{{this.count}}"{{assistant}}
{{#assistant}}Which variable should you choose as the criterion for premises screening?{{assistant}}
{{#assistant}}"Explanation": "{{this.explanation}}"{{assistant}}
{{#assistant}}What are all the premises related to this variable?{{assistant}}
{{#assistant}}"Results": "{{this.Results}}"{{assistant}}
{{~each}}
{{#user}}
—
"determinate premise": "{{determinate premise}}"
"indeterminate premise": "{{indeterminate premise}}"
"topic": "{{topic}}"
"boundary condition": "{{boundary condition}}"
"Inference history": "{{last false history}}"
{{user}}
{{#assistant}}Can you count the cumulative number of times each variable is mentioned by the premises?{{assistant}}
{{#assistant}}"Count": "{{assistant}}
{{#assistant}}{{gen "count" temperature=temperature max tokens=200 stop='
'}}{{assistant}}
{{#assistant}}Which variable should you choose as the criterion for premises screening?{{assistant}}
{{#assistant}}"Explanation": "{{assistant}}
{{#assistant}}{{gen "explanation" temperature=temperature max tokens=200 stop='
'}}{{assistant}}
{{#assistant}}What are all the premises related to this variable?{{assistant}}
{{#assistant}}"Results": "{{assistant}}
{{#assistant}}{{gen "results" temperature=temperature max tokens=200 stop='
'}}{{assistant}}

---

**Prompts used for Premise Exploration**
{{#system}}Suppose you are one of the greatest AI scientists, logicians and mathematicians. Let us think step by step.
Please use Logical Reasoning Rules(LRR) to deduce a "Proposition" from two given "Premises" and the proposition does not include "if".
Logical Reasoning Rules(LRR):
1. "Two premises": "If A,then B. A is true." then "Proposition": "B is true."
2. "Two premises": "If A,then B. B is not true." then "Proposition": "A is not true"
3. "Two premises": "A is either C or D. A is not C." then "Proposition": "A is D."
Please make sure that the "Proposition" is logically correct.
Please make sure that the "Proposition" is not a duplicate of the "Premises".
Please make sure your reasoning is directly deduced from the "Premises" and "Propositions" other than introducing unsourced common knowledge and unsourced information by common sense reasoning.
Please remember that your "Proposition" should be useful to determine whether the "Hypothesis" is True, False or Unknown.
—-
{{system}}

{{#each examples}}
{{#user}}
—
"Premises": "{{this.premises}}"
We want to deduce more propositions to determine the correctness of the following "Hypothesis":
"Hypothesis": "{{this.conclusion}}"

Can you deduce a new "Proposition" from at least two given "Premises"?
{{user}}

{{#assistant}}"Proposition": "{{this.proposition}}"{{assistant}}
{{~each}}
{{#user}}
—
"premises": "{{premises}}"
"boundary condition": "{{boundary condition}}"
We want to derive more propositions to solve the following question:
"question": "{{question}}"
Combined with boundary conditions, can you derive a new "proposition" from at least two given "premises"?
{{user}}

{{#assistant}}"proposition": "{{assistant}}
{{#assistant}}{{gen "proposition" temperature=temperature max tokens=100 stop='
'}}{{assistant}}

---

**Prompts used for Logical Validation**
{{#system}}Suppose you are one of the greatest AI scientists, logicians and mathematicians.
Let us think step by step.
Please use the Logical Reasoning Rules(LRR) to determine whether the deduction of the given "Premises" to a "Proposition" is valid or not, reply with True or False.
Logical Reasoning Rules(LRR):
1. "Two premises": "If A,then B. A is true." then "Proposition": "B is true."
2. "Two premises": "If A,then B. If B,then C." then "Proposition": "If A, then C."
3. "Two premises": "If A,then B. B is not true." then "Proposition": "A is not true"
4. "Two premises": "A is either C or D. A is not C." then "Proposition": "A is D."
—
{{system}}
{{#each examples}}
{{#user}}
—
"Premises": "{{this.premises}}"
"Proposition": "{{this.proposition}}"
{{user}}
{{#assistant}}"Judgement": "Is this deduction valid? {{this.validation}}"{{assistant}}
{{~each}}

---

**Prompts used for Boundary Validation**
{{#system}}Suppose you are one of the greatest AI scientists, logicians, and mathematicians.
Let's think about it step by step.
Answer "True" or "False" to determine whether the existing premises plus a new premise satisfies the boundary condition.
—
{{system}}

{{#each examples}}
{{#user}}
—
"existing premises": "{{this.premises}}"
"new premise": "{{this.new premise}}"
"boundary condition": "{{this.boundary condition}}"

After adding the new premise to the existing premise, does it still meet the boundary conditions?
{{user}}

{{#assistant}}"Judgement": "{{this.judgement}}"{{assistant}}
{{˜each}}

{{#user}}
—
"existing premises": "{{premises}}"
"new premise": "{{proposition}}"
"boundary condition": "{{boundary condition}}"
After adding the new premise to the existing premise, does it still meet the boundary conditions?
{{user}}

{{#assistant}}"Judgement": "{{assistant}}
{{#assistant}}{{select "judgement" options=valid duplicated}}{{assistant}}

---

**Prompts used for Premise Transformation**
{{#system}} Suppose you are one of the greatest AI scientists, logicians, and mathematicians. Let's think about it step by step.
First, please read and analyze the "existing premises", read the definition of transformation;
Transformation: In the one-to-one relationship, when the value of the current variable is determined, it means that this variable can not take other values, and other variables can not take the current value, this reasoning process is transformation.
Check whether relying on a single "premise" and "boundary condition" can translate into other new premises? The new premises should not duplicate any of the existing premises.
If it can be transformed, give the new premises you have deduced; if it can't, answer "None."
Make sure that the new premises you get are helpful in solving the problem.
—
{{system}}
{{#each examples}}
{{#user}}
—
"existing premises": "{{this.premises}}"
"question": "{{this.question}}"
"premise": "{{this.premise}}"
"boundary condition": "{{this.boundary condition}}"
{{user}}
{{#assistant}}Can you derive a new premise based on the premises and boundary condition that help solve the problem?{{assistant}}
{{#assistant}}"new premise": "{{this.new premise}}"{{assistant}}
{{˜each}}

{{#user}}
—
"existing premises": "{{premises}}"
"question": "{{question}}"
"premise": "{{premise}}"
"boundary condition": "{{boundary condition}}"
{{user}}
{{#assistant}}Can you derive a new premise based on the premises and boundary condition that help solve the problem?{{assistant}}

{{#assistant}}"new premise": "{{assistant}}
{{#assistant}}{{gen "premise" temperature=temperature max tokens=50 stop=['
']}}{{assistant}}

---

**Prompts used for Premise Extraction**
{{#system}} Suppose you are one of the greatest AI scientists, logicians, and mathematicians. Let's think about it step by step.
First read and analyze the two sets of definitions defined below;
Premise: A constraint on the absolute position of an object or on the relative relationship between two objects.
Boundary condition: A description of the number of objects and the name of the object.
According to the above definition, summarize the core topics discussed in the following paragraphs and extract the premise and boundary conditions in the context.
—

{{system}}

{{#each examples}}
{{#user}}
—
"context": "{{this.context}}"
{{user}}
{{#assistant}}Can you summarize the core topics of the discussion from the context above?{{assistant}}
{{#assistant}}"topic": "{{this.topic}}"{{assistant}}
{{#assistant}}Can you extract the premise from the context above?{{assistant}}
{{#assistant}}"premise": "{{this.premise}}"{{assistant}}
{{#assistant}}Can you extract the boundary conditions from the context above?{{assistant}}
{{#assistant}}"boundary condition": "{{this.boundary condition}}"{{assistant}}
{{~each}}

{{#user}}
—
"context": "{{context}}"
{{user}}

{{#assistant}}Can you summarize the core topics of the discussion from the context above?{{assistant}}
{{#assistant}}"topic": "{{assistant}}
{{#assistant}}{{gen "topic" temperature=temperature max tokens=50 stop='
'}}{{assistant}}
{{#assistant}}Can you extract the premise from the context above?{{assistant}}
{{#assistant}}"premise": "{{assistant}}
{{#assistant}}{{gen "premise" temperature=temperature max tokens=300 stop=['
n']}}{{assistant}}
{{#assistant}}Can you extract the boundary conditions from the context above?{{assistant}}
{{#assistant}}"boundary condition": "{{assistant}}
{{#assistant}}{{gen "boundary condition" temperature=temperature max tokens=300 stop=['
']}}{{assistant}}

---

**Prompts used for Final Conclusion**
{{#system}}Suppose you are one of the greatest AI scientists, logicians, and mathematicians. Let's think about it step by step.
First read and analyze the "paragraphs" and "questions", then use the "premises", "boundary conditions" and "propositions" to reason which of the options given is the answer to the

"question".
Make sure that your reasoning is derived directly from "premises" and "propositions" rather than introducing passive common sense and passive information through common sense reasoning.
Please note that this is a single choice question.
If you can get the answer directly from the proposition, then you should choose the answer directly, otherwise keep reasoning with the proposition, premises, and boundary conditions until you arrive at a single answer.
—
{{system}}
{{#each examples}}
{{#user}}
—
"context": "{{this.context}}"
"question and options": "{{this.question}}"
{{user}}
{{#assistant}}"Premises": "Let's think step by step, and from the context we can extract these premises: {{this.premises}}"{{assistant}}
{{#assistant}}"Boundary condition": "Let's think step by step, and from the context we can extract these boundary conditions: {{this.boundary condition}}"{{assistant}}
{{#assistant}}"Thoughts": "Let us think step by step. From the premises, we can deduce propositions:{{this.propositions}}"{{assistant}}
{{#assistant}}"Recall the questions and options":"{{this.question}}"{{assistant}}
{{#assistant}}"Reasoning": "Using premises, boundary conditions, and continuing to reason according to the propositions already obtained,{{this.reasoning}}"{{assistant}}
{{#assistant}}"Recall the questions and options":"{{this.question}}"{{assistant}}
{{#assistant}}"Judgement": "Now we know that the answer to this question should be{{this.ans}}"{{assistant}}
{{~each}}
{{#user}}
—
"context": "{{context}}"
"question and options": "{{question}}"
{{user}}
{{#assistant}}"Premises": "Let's think step by step, and from the context we can extract these premises: {{premises}}"{{assistant}}
{{#assistant}}"Boundary condition": "Let's think step by step, and from the context we can extract these boundary conditions: {{boundary condition}}"{{assistant}}
{{#assistant}}"Thoughts": "Let us think step by step. From the premises, we can deduce propositions:{{propositions}}"{{assistant}}
{{#assistant}}"Recall the reasoning history":"{{infer history}}"{{assistant}}
{{#assistant}}"Recall the questions and options":"{{question}}"{{assistant}}
{{#assistant}}"Reasoning": "Using premises, boundary conditions, and continuing to reason according to the propositions already obtained,{{assistant}}
{{#assistant}}{{gen "reasoning" temperature=0.7 max tokens=500 stop=[' ']}}{{assistant}}
{{#assistant}}"Recall the questions and options":"{{question}}"{{assistant}}
{{#assistant}}"Judgement": "Now we know that the answer to this question should be{{assistant}}
{{#assistant}}{{select "judgement" options=choose}}{{assistant}}

