# OpenReview forum: "From Indeterminacy to Determinacy: Augmenting Logical Reasoning Capabilities with Large Language Models"
_ICLR.cc/2024/Conference — ICLR 2024 Conference Withdrawn Submission_

### Official Review · Reviewer_aK68 · 2023-10-28

**Soundness:** 2 fair
**Presentation:** 3 good
**Contribution:** 2 fair
**Rating:** 5
**Confidence:** 4

**Summary:**

This paper presents DetermLR, a CoT-style prompting strategy that elicits stronger reasoning capabilities from LLMs. Specifically, DetermLR iteratively identifies the most promising premises, prioritises and "executes" them, and then stores useful premises in a memory.

Experiments are performed on 4 complex logical reasoning datasets, and DetermLR outperforms the 5 compared baselines, sometimes by a large margin.

**Strengths:**

* CoT-style reasoning is an active and important research area for LLMs as they allow strong reasoning capabilities to be elicited.

* Logical reasoning is an important problem that LLMs are traditionally not strong at. Further investigation in this area is certainly welcome.

* The proposed method achieves good performance on the 5 challenging datasets, outperforming the compared CoT-style prompting strategies.

**Weaknesses:**

* The proposed method is quite simple. Thus, the technical contribution is light. For instance, the "systematic premise identification module" described in Sec. 3.1 is really quite simple, and I don't know whether I'd call it "systematic".

Besides, it closely follows the Cumulative Reasoning (Zhang et al., 2023) technique, with the addition of a memory. Thus, the novelty is limited.

* Some important details have been omitted in the paper, making it harder to understand the technical contributions of the paper. I'll detail it below.

**Questions:**

* In Eq. (1), (2) and (3), what are the definitions of \texttt{relevance}, \texttt{supplement} and \texttt{verify}? If you follow Cumulative Reasoning (CR), are all these functions realised by the LLM?

* In Sec. 3.3, what exactly is the memory?

* What is the definition of "state" in this paper? Is it the number of "invoked" premises?

* The results on CR on FOLIO in your paper are different from that in the original paper (on GPT-4), which is much higher (87.45 vs 69.11). Why such a large discrepancy?

---

> ### Author Response · Authors · 2023-11-18
> **Response to Reviewer aK68 (part I)**
>
> We sincerely thank you for your constructive suggestions and valuable comments! We will answer the questions in detail and will appreciate very much if you could kindly raise your score if your concerns are addressed.
>
>
> Q: *technical contribution*
>
> A: Thanks for your constructive comments! We would like to clarify that considering the critical research question of augmenting the reasoning capabilities of LLMs, we were motivated by three important challenges in bridging the gap between LLM-based reasoning human reasoning. The proposed method provides a solution to this research question in terms of three key factors, which cannot be considered and achieved by baseline methods including CR. Our contributions can be summarized as follows:
>
> 1) We propose to formulate the reasoning process as the transition from indeterminacy to determinacy, which is a novel perspective to describe the essence of solving reasoning tasks;
>
> 2) We employ two-stage quantitative measurements (`relevance` and `supplement`) for premise selection and exploration, which can explore new insights by prioritizing given premises that are conducive to deriving conclusions;
>
> 3) We introduce a reasoning memory module to automate storage and extraction of available premises and reasoning paths, ensuring the consideration of essential historical reasoning details during the iterative reasoning process.
>
> Q: *systematic premise identification*
>
> The implementation of premise identification is to formulate detailed division rules through carefully designed instructions and demonstrations, which are available in revised Appendix. Compared with the identification results, the more important significance of this process is to enable the entire reasoning process to evolve from indeterminacy to determinacy.
> Considering your valuable comments, we will remove the adjective "systematic" to avoid misunderstandings.
>
> Q: *the definitions of \texttt{relevance}, \texttt{supplement}*
>
> A: All these functions are implemented by instructing LLMs and the prompt templates are detailed in revised Appendix.
> We would like to clarify that all three functions are innovations of DetermLR.
> \texttt{relevance} and \texttt{supplement} can prioritize better premises to explore new propositions, unlike randomly selecting premises in CR.
> \texttt{verify} consider multi-aspect verification checks, whereas CR only includes a "logical validity" check.
>
> The main prompts and examples are provided below for easier understanding.
>
> **relevance and supplement**
>
> Prompt:
>
> ```
> Please follow these steps:
> 1.From the determinate premise, select the "Most relevant premise" which has the same subject with "Hypothesis", and give a score from 0 to 1.
> 2.You need to assess how the "Most relevant premise" relates to all the other "determinate premise" and "indeterminate premise", based on Relevance scoring rules.
> 3.The "determinate premise" and "indeterminate premise" with scores higher than 0.25 will be used as the final supplement results, along with Most relevant premise.
> Relevance scoring rules:
> 1. When scoring relevance, 0.25 added for each noun or 0.25 added for each adjective that is the same between two sentences.
> 2. Scores start to accumulate from 0 points, and the upper limit is 1 point.
> 3. If sentence p1 is a hypothetical premise of sentence p2, then add 0.25 to p2. for example: measure "if A then B." and "A is true." Then add 0.25 points to "if A then B".
> ```
>
> E.g.:
>
> ```
> Target: bear is nice and red (True/False/Unknown).
> Determinate premises:
> d1) The bear is big.
> d2) Billy is rough.
> d2) The tiger is rough.
> Indeterminate premises:
> i1) If bear is big, then bear is red.
> i2) If someone is big, then they are nice.
>
> ### After premise priority scoring:
> Most relevant premise $p*$: The bear is big.
> Other Premises:
> i1) If bear is big, then bear is red.(0.75)
> i2) The tiger is rough.(0.0)
> i3) If someone is big, then they are nice.(0.5)
> d2) Billy is rough.(0.0)
> Supplementary premises *$\mathbf{p}_s$*:
> i1) If bear is big, then bear is red.(0.75)
> i3) If someone is big, then they are nice.(0.5)
>
> So the selected premises for exploration: The bear is big. If someone is big, then they are nice. If bear is big, then bear is red.
>
> New proposition $\widehat{p}$: The bear is nice and red.
>
> Conclusion: True.
> ```

---

> > ### Author Response · Authors · 2023-11-18
> > **Response to Reviewer aK68 (part II)**
> >
> > Q: *the definition of \texttt{verify}*
> >
> > A: The main prompts and examples of `verify` are provided below for easier understanding.
> >
> > **1) logical validity**
> >
> > Prompt:
> >
> > ```
> > {{#system}}Suppose you are one of the greatest AI scientists, logicians and mathematicians. Let us think step by step.
> > Please use the Logical Reasoning Rules(LRR) to determine whether the deduction of the given "Premises" to a "Proposition" is valid or not, reply with True or False.
> > Logical Reasoning Rules(LRR):
> > 1. "Two premises": "If A,then B. A is true." then "Proposition": "B is true."
> > 2. "Two premises": "If A,then B. If B,then C." then "Proposition": "If A, then C."
> > 3. "Two premises": "If A,then B. B is not true." then "Proposition": "A is not true"
> > 4. "Two premises": "A is either C or D. A is not C." then "Proposition": "A is D."
> > ----{{/system}}
> > {{~#each examples}}
> > {{#user}}
> > ---
> > "Premises": "{{this.premises}}"
> > "Proposition": "{{this.proposition}}"
> > {{/user}}
> >
> > {{#assistant}}"Judgement": "Is this deduction valid? {{this.validation}}"{{/assistant}}
> > {{~/each}}
> >
> > {{#user}}
> > ---
> > "Premises": "{{premises}}"
> > "Proposition": "{{proposition}}"
> > {{/user}}
> >
> > {{#assistant}}"Judgement": "Is this deduction valid? {{/assistant}}
> > {{#assistant}}{{select "validation" options=valid_validation}}{{/assistant}}
> > ```
> >
> > E.g.:
> >
> > ```
> > Selected premises: All eels are fish. No fish are plants.
> > New proposition: No eels are plants.
> > Validation: True.
> >
> > Selected premises: Nothing that breathes is paper. All animals breathe.
> > New proposition: All animals are paper.',
> > Validation: False.
> > ```
> >
> > **2) useful contribution**
> >
> > Prompt:
> >
> > ```
> > {{#system}}Suppose you are one of the greatest AI scientists, logicians and mathematicians. Let us think step by step.
> > Please determine whether there is a new useful "Proposition". Reply with True or False.
> > ----{{/system}}
> >
> > {{#user}}
> > ---
> > "Proposition": "There is no new proposition that can be deduced from the given premises to determine the correctness of the hypothesis."
> > {{/user}}
> > {{#assistant}}False{{/assistant}}
> >
> > {{#user}}
> > ---
> > "Proposition": "A Czech person wrote a book in 1946."
> > {{/user}}
> > {{#assistant}}True{{/assistant}}
> >
> > {{#user}}
> > ---
> > "Proposition": "There is no new proposition that can be deduced from the given premises that would be useful in determining the correctness of the given hypothesis."
> > {{/user}}
> > {{#assistant}}False{{/assistant}}
> >
> > {{#user}}
> > ---
> > "Proposition": "None of the premises provide information to deduce a proposition related to a Czech person writing a book in 1946."
> > {{/user}}
> > {{#assistant}}False{{/assistant}}
> >
> > {{#user}}
> > ---
> > "Proposition": "{{proposition}}"
> > {{/user}}
> > {{#assistant}}{{select "is_something" options=valid_something}}{{/assistant}}
> > ```
> >
> > E.g.:
> >
> > ```
> > New proposition: No new premises can be derived here.
> > Validation: False.
> >
> > New proposition: I cannot derive new premises.
> > Validation: False.
> > ```
> >
> > **3) duplication avoidance**
> >
> > Prompt:
> >
> > ```
> > {{#system}}Suppose you are one of the greatest AI scientists, logicians and mathematicians. Let us think step by step.
> > Can this "proposition" can be derived using only one "premise"?Please reply with True or False.
> > ----{{/system}}
> >
> > {{~#each examples}}
> > {{#user}}
> > ---
> > "Premises": "{{this.premises}}"
> > "Proposition": "{{this.proposition}}"
> > Can this "proposition" can be derived using only one "premise"?
> > {{/user}}
> >
> > {{#assistant}}"Judgement": "{{this.duplicated}}"{{/assistant}}
> > {{#assistant}}"explanation": "this.explanation"{{/assistant}}
> > {{~/each}}
> >
> > {{#user}}
> > ---
> > "Premises": "{{premises}}"
> > "Proposition": "{{proposition}}"
> > Can this "proposition" can be derived using only one "premise"?
> > {{/user}}
> >
> > {{#assistant}}"Judgement": " {{/assistant}}
> > {{#assistant}}{{select "duplicated" options=valid_duplicated}}{{/assistant}}
> > ```
> >
> > E.g.:
> >
> > ```
> > Existing premises: Miroslav Venhoda was a Czech choral conductor who specialized in the performance of Renaissance and Baroque music. Any choral conductor is a musician. Some musicians love music. Miroslav Venhoda published a book in 1946 called Method of Studying Gregorian Chant.
> >
> > New proposition: If someone is a choral conductor, then he is a musician.
> > Duplicated: True.
> >
> > Existing premises: All eels are fish. No fish are plants. A thing is either a plant or animal. Nothing that breathes is paper. All animals breathe. If a sea eel is either an eel or a plant, then a sea eel is an eel or an animal.
> >
> > New proposition: No eels are plants.
> > Duplicated: False.
> > ```

---

> > > ### Author Response · Authors · 2023-11-18
> > > **Response to Reviewer aK68 (part III)**
> > >
> > > Q: *what exactly is the memory*
> > >
> > > A: As shown in Figure 1, the reasoning memory contains two parts: 1) the updated premise set: previous methods may only use this part for future reasoning; 2) the historical reasoning path: we store the previous successful/failed reasoning steps, which can guide the direction of future premise selection to effectively avoid repeating mistakes and learn from successful experiences like humans.
> > >
> > > For the second part, we provide an example of successful and failed reasoning steps respectively.
> > >
> > > Successful reasoning step:
> > >
> > > `In the NO:x round, we use these “premises”: “If something chases the cat then the cat chases the dog. The bald eagle chases the cat.” and got a “New Determinate Premise”: “The cat chases the dog.”`
> > >
> > > Fail reasoning step:
> > >
> > > `In the NO:y round, we use this "most relevant premise": "Bonnie is either a student who attends the school and is very engaged with school events, or she is not a student who attends the school and is not engaged with school events." and got a "false Proposition": "Bonnie is either a student who attends the school and is very engaged with school events, or she is not a student who attends the school and is not engaged with school events."`
> > >
> > > Q: *the definition of "state"*
> > >
> > > A: In our formulated reasoning process, each reasoning iteration represents a reasoning (visited) state, including all currently available premises. Thus, the number of visited states is equivalent to the required reasoning iteration steps, that is, the total number of newly generated propositions. This definition is consistent with other baseline methods ToT and CR for fair comparison of efficiency.
> > >
> > > Q: *The results on CR on FOLIO in your paper are different from that in the original paper (on GPT-4)*
> > >
> > > A: Thanks for your careful proofreading!
> > > 1) We would like to clarity that we follow the official split "folio-dev" [3] for evaluation, which is the general way for using the FOLIO dataset. The test data used by CR is manually screened based on some standards, the number of premises in their cases and the length of each premise have been significantly reduced, which may make the difficulty easier (Even CoT's accuracy can reach 84.46).
> > >
> > > 2) Consider your concerns, we have added a comparative experiment on "FOLIO wiki" that CR provided. The proposed method DetermLR achieves an accuracy of 88.21, which is still slightly better than CR (87.45).

---

> > > > ### Comment · Reviewer_aK68 · 2023-11-21
> > > >
> > > > Thank you for providing the rebuttal. It partially clarifies some of my questions, including those on the memory and the definitions of relevance, verify, etc. However, my major concerns for the paper, i.e. (1) the limited technical novelty & contribution and (2) the limited improvements over SOTA baselines, still stand. Therefore, I will maintain my original score.

---

### Official Review · Reviewer_KbK2 · 2023-11-01

**Soundness:** 2 fair
**Presentation:** 2 fair
**Contribution:** 3 good
**Rating:** 5
**Confidence:** 4

**Summary:**

The paper introduces a new reasoning framework, DetermLR, aimed at enhancing the logical reasoning capabilities of large language models. The study addresses challenges of LLMs facing in emulating human-like reasoning, including selecting appropriate reasoning structures, efficiently using known information, and incorporating past reasoning into future decisions. DetermLR use premise identification, premise prioritization and exploration, and an iterative process with reasoning memory. Experimental results on logical reasoning tasks show that DetermLR outperforms baselines in terms of reasoning performance and efficiency.

**Strengths:**

- This paper is well-motivated and proposes a novel framework that tackles the challenges in emulating human-like logical reasoning.
- The method incorporates a prioritized strategy to direct the reasoning process; history reasoning information including valid and invalid intermediate results to continue reasoning, which are key components for effective reasoning.
- The experimental results demonstrate the effectiveness and efficiency of the proposed framework compared to baseline methods.

**Weaknesses:**

1. The paper lacks much necessary information about framework.
- How do you score relevance and supplement? A transparent LM or directly instruct GPT-4?
- From Fig.1 first step, the authors seem to filter out determinate premises by words matching (eg, the sentence including Gary). However, in some cases, the relationship between premises and conclusion is only logic-level. eg., If Erin is round, then Gary is quiet.
- How do you exploit history reasoning paths information?


2. [1] paper use a similar idea, alternating between premises selection and inference, despite of lackness of history reasoning paths. But the authors do not emphasis the point in details. In facts, I think those failure cases can help reduce search space.

[1] Creswell, Antonia, Murray Shanahan, and Irina Higgins. "Selection-inference: Exploiting large language models for interpretable logical reasoning." arXiv preprint arXiv:2205.09712 (2022).

**Questions:**

See weakness above. If the authors can perfectly solve my issues, I will consider improving my score.

---

> ### Author Response · Authors · 2023-11-18
> **Response to Reviewer KbK2 (part I)**
>
> We sincerely thank you for your constructive suggestions and valuable comments! We will try our best to solve your issues and will appreciate very much if you could kindly raise your score if your concerns are addressed.
>
> Q: *lacks much necessary information about framework*
>
> A: Thanks for your reminding! We have reorganized the structure of Method Section. We add a Problem Formulation Subsection to provide more explanations and revise the notations to make the implementation easier to understand.
>
> Q: *how to score relevance and supplement*
>
> A: We implement the relevance and supplement scorer by prompting LLMs. We provide prompt templates and an example below for easier understanding.
>
> Prompt:
> ```
> Please follow these steps:
> 1.From the determinate premise, select the "Most relevant premise" which has the same subject with "Hypothesis", and give a score from 0 to 1.
> 2.You need to assess how the "Most relevant premise" relates to all the other "determinate premise" and "indeterminate premise", based on Relevance scoring rules.
> 3.The "determinate premise" and "indeterminate premise" with scores higher than 0.25 will be used as the final supplement results, along with Most relevant premise.
> Relevance scoring rules:
> 1. When scoring relevance, 0.25 added for each noun or 0.25 added for each adjective that is the same between two sentences.
> 2. Scores start to accumulate from 0 points, and the upper limit is 1 point.
> 3. If sentence p1 is a hypothetical premise of sentence p2, then add 0.25 to p2. for example: measure "if A then B." and "A is true." Then add 0.25 points to "if A then B".
> ```
>
> E.g.:
>
> ```
> Target: bear is nice and red (True/False/Unknown).
> Determinate premises:
> d1) The bear is big.
> d2) Billy is rough.
> d2) The tiger is rough.
> Indeterminate premises:
> i1) If bear is big, then bear is red.
> i2) If someone is big, then they are nice.
>
> ### After premise priority scoring:
> Most relevant premise $p*$: The bear is big.
> Other Premises:
> i1) If bear is big, then bear is red.(0.75)
> i2) The tiger is rough.(0.0)
> i3) If someone is big, then they are nice.(0.5)
> d2) Billy is rough.(0.0)
> Supplementary premises *$\mathbf{p}_s$*:
> i1) If bear is big, then bear is red.(0.75)
> i3) If someone is big, then they are nice.(0.5)
>
> So the selected premises for exploration: The bear is big. If someone is big, then they are nice. If bear is big, then bear is red.
>
> New proposition $\widehat{p}$: The bear is nice and red.
>
> Conclusion: True.
> ```
>
> Q: *how to filter out determinate premises*
>
> A:
> 1) Determinate premises are defined as simple statements directly related to the target conclusion, requiring to instruct LLMs to understand the target rather than simply filtering it by word matching. Taking Case B in our paper as an example, the determinate premise is like "Someone is male/female bachelor/PhD degree", the partial solution to the problem.
>
> 2) For your given example, the premise **If Erin is round, then Gary is quiet** (denoted as $i_7$) is identified as indeterminate premises since its "if-then" structure cannot directly derive new determinate facts without interacting with other premises. But different indeterminate premises also have different priorities for deriving conclusions, which is exactly the meaning of supplement scoring in the second part of our premise prioritization. Since the target revolves around "Gary" and "round", $i_7$ may have a higher supplementary score than other premises without "Gary", and is more likely to be used to generate new propositions to close to the target.
>
> Q: *how to exploit history reasoning paths information*
>
> A: The historical reasoning path is mainly used at the beginning of each reasoning iteration. We use memory extraction to merge the successful and failed reasoning steps of previous iterations.
>
> Successful reasoning step:
>
> `In the NO:x round, we use these “premises”: “If something chases the cat then the cat chases the dog. The bald eagle chases the cat.” and got a “New Determinate Premise”: “The cat chases the dog.”`
>
> Failed reasoning step:
>
> `In the NO:y round, we use this "most relevant premise": "Bonnie is either a student who attends the school and is very engaged with school events, or she is not a student who attends the school and is not engaged with school events." and got a "false Proposition": "Bonnie is either a student who attends the school and is very engaged with school events, or she is not a student who attends the school and is not engaged with school events."`

---

> > ### Author Response · Authors · 2023-11-18
> > **Response to Reviewer KbK2 (part II)**
> >
> > Q: *compared to Selection-inference*
> >
> > A: Thanks for your constructive comments! We would like to clarify that compared to the Selection-inference (SI) framework, our DetermLR has three main innovations:
> >
> > 1) We propose to formulate the reasoning process as the transition from indeterminacy to determinacy, which is a novel perspective to describe the essence of solving reasoning tasks compared to SI;
> >
> > 2) We employ two-stage quantitative measurements (`relevance` and `supplement`) for premise selection and exploration, which is more nuanced than SI to explore new insights by prioritizing given premises that are conducive to deriving conclusions;
> >
> > 3) We introduce a reasoning memory module to automate storage and extraction of available premises and reasoning paths, ensuring the consideration of essential historical reasoning details during the iterative reasoning process.
> >
> > Considering your valuable comments, we have added the discussion of Selection-Inference in our revised paper.

---

> ### Comment · Reviewer_KbK2 · 2023-11-23
> **Re' Response to Reviewer KbK2**
>
> Thank you for your time and efforts in providing the rebuttal. I appreciate the response to the framework details. For the comparison with SI, I cannot see substantial differences. Specifically,
>
> > We propose to formulate the reasoning process as the transition from indeterminacy to determinacy
>
> The perspective is new. However, the reasoning process is pretty much similar.
>
> > We employ two-stage quantitative measurements (`relevance` and `supplement`) for premise selection and exploration
>
> This part of the contribution is acknowledged.
>
> > We introduce a reasoning memory module to automate storage and extraction of available premises and reasoning paths
>
> This is also a widely applied trick in many frameworks, such as langchain, babyAGI, etc.
>
> In summary, I stand with my concerns on the incremental novelty of this work.

---

### Official Review · Reviewer_2L5z · 2023-11-04

**Soundness:** 3 good
**Presentation:** 2 fair
**Contribution:** 2 fair
**Rating:** 5
**Confidence:** 3

**Summary:**

The paper proposes a new prompting technique for logic reasoning, DetermLR, which first classifies the given premises into determinate and indeterminate, and then sorts them by priority. Then DetermLR will first explore the premises with high priority and store the new conclusions into the memory for future reference. The proposed method shows superior results on LogiQA, ProofWriter, FOLIO, and LogicalDeduction, over multiple baselines.

**Strengths:**

1. The proposed method seems to be quite effective on four logic reasoning datasets, compared to multiple baselines.
2. The paper is mostly clear and well-written.

**Weaknesses:**

1. It is not clear how each module in the proposed framework is implemented. Such information is completely missing in the method section while judging from the experiment section, it seems that all of them are implemented by prompting GPT4. However, it is still unclear how the scorers are implemented and what is the threshold $\theta$ for supplementary premises filtering.
2. I'm not quite sure why the proposed method can select a reasoning structure. The main technique of the proposed method seems to be classifying the given premises into determinate and indeterminate, and sorting the premises by priority. It seems to adopt a mostly linear reasoning structure with memory reference.

I'm willing to increase my score if my concerns are properly addressed.

**Questions:**

1. How is the time/compute efficiency of the proposed method compared to the baselines? I understand that the proposed method visited fewer states during inference, but I'm not quite sure if the compute/number of GPT4 prompting for each state visiting is the same as the baselines.

---

> ### Author Response · Authors · 2023-11-18
> **Response to Reviewer 2L5z**
>
> We sincerely thank you for your constructive suggestions and valuable comments! We will try our best tackle your concerns and will appreciate very much if you could kindly raise your score.
>
> Q: *how each module in the proposed framework is implemented, how the scorers are implemented and what is the threshold for supplementary premises filtering*
>
> A: Thanks for your reminding!
> 1) The essential process of the proposed method including premise identification, premise prioritization, verification checking, and reasoning memory are all implemented by prompting LLMs with carefully designed instructions and demonstrations.
>
> 2) For premise priority scoring, the prompt templates and examples are included as follows, with the threshold set to 0.25. More prompt templates are available in revised Appendix.
>
> Prompt:
>
> ```
> Please follow these steps:
> 1.From the determinate premise, select the "Most relevant premise" which has the same subject with "Hypothesis", and give a score from 0 to 1.
> 2.You need to assess how the "Most relevant premise" relates to all the other "determinate premise" and "indeterminate premise", based on Relevance scoring rules.
> 3.The "determinate premise" and "indeterminate premise" with scores higher than 0.25 will be used as the final supplement results, along with Most relevant premise.
> Relevance scoring rules:
> 1. When scoring relevance, 0.25 added for each noun or 0.25 added for each adjective that is the same between two sentences.
> 2. Scores start to accumulate from 0 points, and the upper limit is 1 point.
> 3. If sentence p1 is a hypothetical premise of sentence p2, then add 0.25 to p2. for example: measure "if A then B." and "A is true." Then add 0.25 points to "if A then B".
> ```
>
> E.g.:
>
> ```
> Target: bear is nice and red (True/False/Unknown).
> Determinate premises:
> d1) The bear is big.
> d2) Billy is rough.
> d2) The tiger is rough.
> Indeterminate premises:
> i1) If bear is big, then bear is red.
> i2) If someone is big, then they are nice.
>
> ### After premise priority scoring:
> Most relevant premise $p*$: The bear is big.
> Other Premises:
> i1) If bear is big, then bear is red.(0.75)
> i2) The tiger is rough.(0.0)
> i3) If someone is big, then they are nice.(0.5)
> d2) Billy is rough.(0.0)
> Supplementary premises *$\mathbf{p}_s$*:
> i1) If bear is big, then bear is red.(0.75)
> i3) If someone is big, then they are nice.(0.5)
>
> So the selected premises for exploration: The bear is big. If someone is big, then they are nice. If bear is big, then bear is red.
>
> New proposition $\widehat{p}$: The bear is nice and red.
>
> Conclusion: True.
> ```
>
> Q: *why the proposed method can select a reasoning structure*
>
> A: Thanks for your insightful comments!
> We would like to clarify that we does not focus on pre-defining the reasoning structure, while the structure can be derived by reviewing the connections between the corresponding states of each reasoning iteration.
> Pre-defining the reasoning structure before solving problems is quite different from the way of human reasoning. Humans actually do not preset a structure when facing a reasoning problem, while the so-called reasoning structure should be formed based on the reviewed reasoning results after solving the problem.
> Our experiments also show that compared to predefined thinking structures, the complexity of the reasoning structure reviewed by our method is more consistent with the problem difficulty itself.
>
> Q: *efficiency of the proposed method compared to the baselines*
>
> A: Thanks for your constructive suggestions! We have added complexity analysis for more detailed efficiency comparisons. We choose ToT [1] and CR [2] as baselines and randomly select 100 cases of FOLIO to compute the average inference time for each visited state.
> The results below show that since our DetermLR can substantially save average states with slight increment on inference time per state, the overall inference efficiency of DetermLR is still better than ToT and CR.
>
> |Method|# states per case|Inference time per state|Inference time per case|
> |-|:-:|:-:|:-:|
> |ToT|18.40|7.77s|142.93s|
> |CR|14.51|6.86s|99.69s|
> |**DetermLR**|**7.69**|8.05s|**61.98s**|

---

### Official Review · Reviewer_96xE · 2023-11-07

**Soundness:** 2 fair
**Presentation:** 2 fair
**Contribution:** 3 good
**Rating:** 3
**Confidence:** 2

**Summary:**

They propose an approach for improving the logical reasoning of LLMs. Their approach is to structure the prompt with three main components including their so-called, premise identification, premise prioritization, and iterative process with reasoning memory.  They conduct experiments on four logical reasoning datasets using their prompting strategy. Their approach outperforms other recent strategies for prompting LLMs such as chain-of-though and some other variations.

**Strengths:**

-The authors propose a new strategy for structuring the prompt for LLM to make them perform logical reasoning with a higher accuracy.
-The experiments show the effectiveness of the proposed approach compared to the existing structured prompting strategies.

**Weaknesses:**

-The terminology, notations, and in general the explanation of the proposed approach was not very clear to me.
-The results focused on the selected subsets of datasets -selected by authors. No comparison with other results on these datasets [outside this work] was made. Or this was not made explicit at least in the paper as far as I understood.
-The results were reported only on GPT4.

See some details in the Questions section.

**Questions:**

-From the provided examples, it was not made clear to me why the term indeterminacy was chosen for some parts of the information.
-in section 3.1., the authors explained the premise identification, and then only in section 3.2 they started introducing formal notations.
-The formalization and notation are somewhat superficial and not really used to help understanding.
-The flow of information and how these modules exactly work is not clear, are you expecting the LLM to do these steps with a few shots of in-context learning? For example, how the model was asked to identify the premises in the first step? These are just very hard to read from the current presentation of the paper.
--Not clear what the authors mean by verification check? how the verification is performed, and what is the kind of computation used here for verification?
--What do you mean by states and visited states? I did not see this defined in the paper.

---

> ### Author Response · Authors · 2023-11-18
> **Response to Reviewer 96xE (part I)**
>
> We sincerely thank you for your constructive suggestions and valuable comments! We will answer the questions in detail and will appreciate very much if you could kindly raise your score if your concerns are addressed.
>
> Q: *terminology, notations, and in general the explanation of the proposed approach*
>
> A: Thanks for your reminding!
> We have reorganized the method section to make notations clearer and add more explanations in the revision.
>
> Q: *results in selected datasets*
>
> A: We would like to clarify that the test data used in our comparative experiments is mostly based on the official repositories without any further post-preprocessing [1-3], which is also the most common way of using these datasets. For LogiQA, since it contains different types of examination questions, non-logical reasoning like common sense reasoning is not our focus. In order to accurately evaluate the ability of LLMs to perform logical reasoning using only given conditions without relying on any external knowledge, we carefully reviewed LogiQA, excluded cases with unclear conditional expressions, and finally remained 179 high-quality questions as a curated collection. With our effort of data proofreading, the processed data will be released to the community to facilitate further studies.
>
> Q: *The results were reported only on GPT4.*
>
> A: Thanks for your valuable comments! We agree that it is indeed necessary to add more base models to further compare the effectiveness of the proposed method.
> We plan to add GPT-3.5-turbo and LLaMA2 as engines for comparison. Currently we can provide the results of GPT-3.5-turbo on ProofWriter and FOLIO below, showing that all methods perform uniformly worse than GPT-4 (by about 0.1 accuracy). But DetermLR still outperforms all baselines on the same base model. We will complete these results and more discussions about the impact of base models soon in the future revision.
> |Method|ProofWriter|FOLIO|
> |-|:-:|:-:|
> |Standard|49.51|36.17|
> |CoT|54.41|45.00|
> |CoT-SC|57.34|48.67|
> |ToT|59.80|54.17|
> |CR|59.80|59.17|
> |DetermLR|**63.72**|**68.83**|
>
> Q: *why the term indeterminacy was chosen for some parts of the information*
>
> A: Thanks for your insightful comments! We will introduce in detail the necessity of the definitions of "indeterminacy" and "determinancy", and an example of the reasoning process from indeterminacy to determinancy.
>
> 1) We introduce indeterminacy and determinancy to distinguish whether the information in a given premise is clear for deriving the conclusion. Only simple statements directly related to the conclusion are treated as determinate premises. So "indeterminacy" can indicate that those premises must be combined with other conditions to deduce new determinate information by removing the indeterminate structures (if-then, or).
>
> 2) Taking Case B in our paper as an example, all original given premises are treated as indeterminate premises based on our definition. In this case, we just need to **further infer each original premise to obtain potentially determinate information**. From **"$i_2$: The educational levels of A, B, and C are the same, while those of F and G are different"**, we know that: **there are at least 4 people with education levels of A, B, and C.** Then we can derive a determinate premise: **"$d_1$: A, B, and C are Bachelors."** based on the boundary condition: There are only 3 PhDs among the 7 people. Then more determinate premises can be deduced in the subsequent reasoning process. Therefore, it is consistent with our assumptions about the reasoning process: from indeterminacy to determinacy.
>
>
> [1] https://allenai.org/data/proofwriter
>
> [2] https://github.com/Yale-LILY/FOLIO
>
> [3] https://github.com/suzgunmirac/BIG-Bench-Hard/blob/main/bbh/logical_deduction_seven_objects.json

---

> > ### Author Response · Authors · 2023-11-18
> > **Response to Reviewer 96xE (part II)**
> >
> > Q: *how these modules exactly work and how the model was asked to identify the premises in the first step*
> >
> > A: Thanks for your reminding!
> > 1) We would like to clarify that the process of premise identification, premise prioritization, and reasoning memory are all implemented by prompting LLMs with carefully designed instructions and demonstrations.
> > The essential prompt templates are available in revised Appendix.
> >
> > 2) Specifically for premise identification, we include the prompt below.
> > ```
> > {{#system}} Suppose you are one of the greatest AI scientists, logicians and mathematicians.
> > Let us think step by step.
> > First, read and analyze the following definition:
> > Determinate premise: The premise contains the same noun or adjective as the Hypothesis, and the premise is not in the structure of ``if...'' or ``if...then...''.
> > Second, read and analyze the ``Premise'' and ``Hypothesis'' .Judge ``Premise'' is ``determinate premise'' or not.
> > Third, please make sure your classification decisions are derived directly from definitions, rather than unsourced common sense.
> > ---
> > {{/system}}
> > {{~#each examples}}
> > {{#user}}
> > ---
> > ``Premise'': ``{{this.Premise}}''
> > ``Hypothesis'': ``{{this.Hypothesis}}''
> > {{/user}}
> > \raggedright{{#assistant}}``Judgement'':``Is this ''Premise`` a ''determinate premise`` or not?
> > {{this.usefulness}}'' {{/assistant}}
> > {{#assistant}}``Explanation'': {{this.Explanation}}{{/assistant}}
> > {{~/each}}
> > ```
> > Here is an example that since the target revolves around **bear** and **rough**, the simple statements including these key entities are identified as determinate premises (d1-d3), while others containing "if-then" structures are indeterminate premises.
> >
> > ```
> > Input target: bear is rough. (True/False/Unknown)
> > Input Premises: 1) The bear is big. 2) Billy is rough. 3) The tiger is rough. 4) If bear is big, then bear is red. 5) If someone is big, then they are nice.
> >
> > Determinate premises: d1) The bear is big. d2) Billy is rough. d3) The tiger is rough.
> > Indeterminate premises: i1) If bear is big, then bear is red. i2) If someone is big, then they are nice.
> > ```
> >
> > Q: *visited states*
> >
> > A: In our formulated reasoning process, each reasoning iteration represents a (visited) state as an attempt to reason about a new proposition. Thus, the number of visited states is equivalent to the required reasoning iteration steps, that is, the total number of newly generated  propositions. This definition is consistent with baseline methods (Tree-of-Thoughts [4] and Cumulative-Reasoning [5]) for fair comparison of efficiency.
> >
> > [4] Yao, Shunyu, et al. "Tree of thoughts: Deliberate problem solving with large language models."
> >
> > [5] Zhang, Yifan, et al. "Cumulative reasoning with large language models."

---

> > > ### Author Response · Authors · 2023-11-18
> > > **Response to Reviewer 96xE (part III)**
> > >
> > > Q: *verification check*
> > >
> > > A: Thanks for your valuable comments!
> > > When a new proposition is generated by selected premises, the purpose of "verification check" is to evaluate the quality of the proposition to determine whether to update it into the current premise set.
> > > We prompt LLMs to assess the new proposition whether satisfy the requirements for conclusion derivation in multiple aspects: "logical validity", "useful contribution", and "duplication avoidance".
> > > The prompt templates and examples are detailed as follows.
> > >
> > > **1) logical validity**
> > >
> > > Prompt:
> > >
> > > ```
> > > {{#system}}Suppose you are one of the greatest AI scientists, logicians and mathematicians. Let us think step by step.
> > > Please use the Logical Reasoning Rules(LRR) to determine whether the deduction of the given "Premises" to a "Proposition" is valid or not, reply with True or False.
> > > Logical Reasoning Rules(LRR):
> > > 1. "Two premises": "If A,then B. A is true." then "Proposition": "B is true."
> > > 2. "Two premises": "If A,then B. If B,then C." then "Proposition": "If A, then C."
> > > 3. "Two premises": "If A,then B. B is not true." then "Proposition": "A is not true"
> > > 4. "Two premises": "A is either C or D. A is not C." then "Proposition": "A is D."
> > > ----{{/system}}
> > > {{~#each examples}}
> > > {{#user}}
> > > ---
> > > "Premises": "{{this.premises}}"
> > > "Proposition": "{{this.proposition}}"
> > > {{/user}}
> > >
> > > {{#assistant}}"Judgement": "Is this deduction valid? {{this.validation}}"{{/assistant}}
> > > {{~/each}}
> > >
> > > {{#user}}
> > > ---
> > > "Premises": "{{premises}}"
> > > "Proposition": "{{proposition}}"
> > > {{/user}}
> > >
> > > {{#assistant}}"Judgement": "Is this deduction valid? {{/assistant}}
> > > {{#assistant}}{{select "validation" options=valid_validation}}{{/assistant}}
> > > ```
> > >
> > > E.g.:
> > >
> > > ```
> > > Selected premises: All eels are fish. No fish are plants.
> > > New proposition: No eels are plants.
> > > Validation: True.
> > >
> > > Selected premises: Nothing that breathes is paper. All animals breathe.
> > > New proposition: All animals are paper.',
> > > Validation: False.
> > > ```
> > >
> > > **2) useful contribution**
> > >
> > > Prompt:
> > >
> > > ```
> > > {{#system}}Suppose you are one of the greatest AI scientists, logicians and mathematicians. Let us think step by step.
> > > Please determine whether there is a new useful "Proposition". Reply with True or False.
> > > ----{{/system}}
> > >
> > > {{#user}}
> > > ---
> > > "Proposition": "There is no new proposition that can be deduced from the given premises to determine the correctness of the hypothesis."
> > > {{/user}}
> > > {{#assistant}}False{{/assistant}}
> > >
> > > {{#user}}
> > > ---
> > > "Proposition": "A Czech person wrote a book in 1946."
> > > {{/user}}
> > > {{#assistant}}True{{/assistant}}
> > >
> > > {{#user}}
> > > ---
> > > "Proposition": "There is no new proposition that can be deduced from the given premises that would be useful in determining the correctness of the given hypothesis."
> > > {{/user}}
> > > {{#assistant}}False{{/assistant}}
> > >
> > > {{#user}}
> > > ---
> > > "Proposition": "None of the premises provide information to deduce a proposition related to a Czech person writing a book in 1946."
> > > {{/user}}
> > > {{#assistant}}False{{/assistant}}
> > >
> > > {{#user}}
> > > ---
> > > "Proposition": "{{proposition}}"
> > > {{/user}}
> > > {{#assistant}}{{select "is_something" options=valid_something}}{{/assistant}}
> > > ```
> > >
> > > E.g.:
> > >
> > > ```
> > > New proposition: No new premises can be derived here.
> > > Validation: False.
> > >
> > > New proposition: I cannot derive new premises.
> > > Validation: False.
> > > ```
> > >
> > > **3) duplication avoidance**
> > >
> > > Prompt:
> > >
> > > ```
> > > {{#system}}Suppose you are one of the greatest AI scientists, logicians and mathematicians. Let us think step by step.
> > > Can this "proposition" can be derived using only one "premise"?Please reply with True or False.
> > > ----{{/system}}
> > >
> > > {{~#each examples}}
> > > {{#user}}
> > > ---
> > > "Premises": "{{this.premises}}"
> > > "Proposition": "{{this.proposition}}"
> > > Can this "proposition" can be derived using only one "premise"?
> > > {{/user}}
> > >
> > > {{#assistant}}"Judgement": "{{this.duplicated}}"{{/assistant}}
> > > {{#assistant}}"explanation": "this.explanation"{{/assistant}}
> > > {{~/each}}
> > >
> > > {{#user}}
> > > ---
> > > "Premises": "{{premises}}"
> > > "Proposition": "{{proposition}}"
> > > Can this "proposition" can be derived using only one "premise"?
> > > {{/user}}
> > >
> > > {{#assistant}}"Judgement": " {{/assistant}}
> > > {{#assistant}}{{select "duplicated" options=valid_duplicated}}{{/assistant}}
> > > ```
> > >
> > > E.g.:
> > >
> > > ```
> > > Existing premises: Miroslav Venhoda was a Czech choral conductor who specialized in the performance of Renaissance and Baroque music. Any choral conductor is a musician. Some musicians love music. Miroslav Venhoda published a book in 1946 called Method of Studying Gregorian Chant.
> > >
> > > New proposition: If someone is a choral conductor, then he is a musician.
> > > Duplicated: True.
> > >
> > > Existing premises: All eels are fish. No fish are plants. A thing is either a plant or animal. Nothing that breathes is paper. All animals breathe. If a sea eel is either an eel or a plant, then a sea eel is an eel or an animal.
> > >
> > > New proposition: No eels are plants.
> > > Duplicated: False.

---

> > > > ### Comment · Reviewer_96xE · 2023-11-22
> > > >
> > > > Thank you for clarifying my questions.  This was helpful. However, the amount of missing details and the addition of new promised results as well as the conducted ones during rebuttal plus integrating all in the paper is a major revision. I think the paper needs a whole new version to be accepted.

---

### Author Response · Authors · 2023-11-18
**Quick Clarification**

We sincerely thank all reviewers for the thoughtful comments and valuable suggestions! We notice that reviewers have similar concerns regarding the implementation of the proposed method and notation/presentation in the method introduction. We would like to make a quick clarification for these two aspects, hoping to clear up any doubts and possible misunderstandings.

1) We would like to clarify that the essential process of the proposed method including premise identification, premise prioritization, verification checking, and reasoning memory are all implemented by prompting LLMs with carefully designed instructions and demonstrations. We have included prompts and examples in the specific questions raised by each reviewer. More details prompt templates are available in revised Appendix.

2) Thank you for your careful proofreading and reminding on the paper presentation! We have considered these valuable comments and reorganized the structure of Method Section. To make the logical reasoning task clearer, we add a Problem Formulation Subsection to provide more explanations and revise the notations to make the implementation of each module easier to understand.